# Deterministic early endosomal maturations emerge from a stochastic trigger-and-convert mechanism

Harrison M. York [1,8] ✉, Kunaal Joshi [2,8], Charles S. Wright [2,8], Laura Z. Kreplin [1], Samuel J. Rodgers [3], Ullhas K. Moorthi[1], Hetvi Gandhi[1], Abhishek Patil [1], Christina A. Mitchell [3], Srividya Iyer-Biswas [2,4] ✉ & Senthil Arumugam [1,5,6,7] ✉

Endosomal maturation is critical for robust and timely cargo transport to specific cellular compartments. The most prominent model of early endosomal maturation involves a phosphoinositide-driven gain or loss of specific proteins on individual endosomes, emphasising an autonomous and stochastic description. However, limitations in fast, volumetric imaging long hindered direct whole cell-level measurements of absolute numbers of maturation events. Here, we use lattice light-sheet imaging and bespoke automated analysis to track individual very early (APPL1-positive) and early (EEA1-positive) endosomes over the entire population, demonstrating that direct inter-endosomal contact drives maturation between these populations. Using fluorescence lifetime, we show that this endosomal interaction is underpinned by asymmetric binding of EEA1 to very early and early endosomes through its N- and C-termini, respectively. In combination with agent-based simulation which supports a 'trigger-and-convert' model, our findings indicate that APPL1- to EEA1-positive maturation is driven not by autonomous events but by heterotypic EEA1-mediated interactions, providing a mechanism for temporal and population-level control of maturation.

In cellular signal transduction, information is often encoded as a transient pulse or as a temporal pattern of signals. The binding of growth factors to their receptors results in the activation of secondary messengers followed by critical deactivation of receptors through the interaction with phosphatases and lysosomal degradation[1,2]. This combination of events in the signal transduction pathway typically encodes the temporal pattern. The endosomal pathway, where both spatial trafficking and biochemical maturation of endosomes occur in parallel, is a central process that modulates the interaction of receptors with enzymes embedded in other organelles, such as the endoplasmic reticulum, or degradation via the lysosomal pathway[3]. Following formation at the plasma membrane via endocytosis, endosomes carrying cargoes undergo maturation processes[4] facilitated by the concerted effects of motility, inter-endosomal fusions, fissions,

[1]Monash Biomedicine Discovery Institute, Faculty of Medicine, Nursing and Health Sciences, Monash University, Clayton/Melbourne, VIC 3800, Australia. [2]Department of Physics and Astronomy, Purdue University, West Lafayette, IN 47907, USA. [3]Department of Biochemistry and Molecular Biology, Biomedicine Discovery Institute, Monash University, Clayton/Melbourne, VIC 3800, Australia. [4]Santa Fe Institute, Santa Fe, NM 87501, USA. [5]ARC Centre of Excellence in Advanced Molecular Imaging, Monash University, Clayton/Melbourne, VIC 3800, Australia. [6]European Molecular Biological Laboratory Australia (EMBL Australia), Monash University, Clayton/Melbourne, VIC 3800, Australia. [7]Single Molecule Science, University of New South Wales, Sydney, NSW 2052, Australia. [8]These authors contributed equally: Harrison M. York, Kunaal Joshi, Charles S. Wright. ✉e-mail: harrison.york@monash.edu; iyerbiswas@purdue.edu; senthil.arumugam@monash.edu

and endosomal conversions. These latter switch-like processes involve protein conversions, in which one specific set of proteins is shed and another acquired[5,6]. This occurs in concert with phosphoinositide conversions, in which specific phosphoinositide species act as the modules of coincidence detection[7]. Thus, phosphoinositide species provide a second layer of regulation, governing which proteins will localise to a specific subset of endosomes[8,9]. Epidermal growth factor receptors (EGFR) have been shown to depend on dynein for receptor sorting and localisation to mature endosomes[10,11]. In addition, localisation of EGFR to EEA1 compartments was delayed when dynein was inhibited. On the other hand, the expansion of APPL1 compartments enhanced EGFR signalling, consistent with the role of endosomal maturation in the modulating temporal activity of receptors in endosomes. An open question that arises then is, how does motility or subcellular localisation influence endosomal maturation? Furthermore, in the context of trafficking of cargo such as EGFR, that respond to pulsatile patterns of ligands, how do populations of endosomes mature in a timely manner that ensures accurate signal interpretation?

Our current understanding of the dynamics of endosomal maturations comes from seminal live-cell imaging studies that captured the process of individual endosomes undergoing direct conversions[5,6]. These observations led to the prevailing single endosome-centric model wherein a phosphoinositide switch controls the transition from adaptor protein, phosphotyrosine interacting with PH domain and leucine zipper 1 (APPL1) to early endosomal antigen 1 (EEA1) on an individual endosome[6]. APPL1 binds to endosomes via its PH and PTB domains with phosphoinositides. At the level of early endosomes, it putatively binds via a coincidence detection mechanism of PI(3,4)P2 and Rab5[12]. EEA1 binds to endosomes via its interaction with Rab5 and PI(3)P-binding FYVE domain at its C-terminus[13] as well as its N-terminal interaction with Rab5[14]. Zoncu et al. showed that PI(3)P was required for long-lived EEA1 endosomes; they also observed reversions of EEA1-to-APPL1 conversion upon inducible depletion of PI(3)P, suggesting that APPL1 to EEA1 maturation is underpinned by a phosphoinositide switch resulting in PI(3)P production[6]. In mammals, PI(3,4)P2 can be dephosphorylated to PI(3)P by either of two phosphoinositide 4-phosphatases, INPP4A and INPP4B[15,16], which have been suggested to have distinct intracellular localisations, with INPP4A being found on Rab5-positive endosomes[8,17]. Nonetheless, these single endosome-centric maturation models do not address population-level maturation rates, which are essential for bulk regulation of receptor trafficking, and therefore signal interpretation. Secondly, the single endosome-centric models rely on the stochastic binding of molecules, which is unpredictable as a mechanism. Stochasticity poses crucial challenges in maintaining causal ordering and temporal specificity, i.e., a tight probability distribution of events in time. However, despite the emphasis on stochasticity in constituent dynamics in the vesicular transport system[18–21], endosomal trafficking processes display an extraordinary degree of robustness and predictability in delivering cargo to specific intracellular destinations, and receptors transported through the endosomal system show reproducible signalling outcomes. These properties suggest that there exist mechanisms to counter the stochasticity of the constituent processes and thus achieve tight control over the maturation, trafficking, and dynamics of the intracellular transport system. A limiting factor in extending and reconciling the previously established single endosome-centric model to population-level maturation rates has been the difficulty in directly measuring these dynamic events at whole cell levels.

Here, we use lattice light-sheet microscopy (LLSM) live-cell imaging, which allows rapid imaging of whole cell volumes for extended periods of time[22], to measure the whole cell dynamics of APPL1 and EEA1. To quantify these data, we develop a bespoke endosome detection and tracking algorithm to measure large numbers of endosomal collisions, fusions, and conversions occurring within many single cells over a prolonged period of imaging. We complemented these methods with live-cell fluorescence lifetime microscopy (FLIM) to interrogate the molecular orientation of EEA1, a head-to-head homodimer bound to maturing early endosomes. We show that very early endosome (VEE) to early endosome (EE) conversion is a multistep process, underpinned by the multiple asymmetric binding sites of EEA1 and its cyclical conformation changes, which is brought about by endosomal collisions and heterotypic fusions. Through simulations, we test the effectiveness of our proposed mechanism in predicting the maturation time course, specifically, the conversion from APPL1 to EEA1 and from N- to C-terminal EEA1 attachments. These results warrant a significant upgrade to the model of endosomal maturations, with heterotypic interactions—where collisions lead to triggered conversions or fusions—forming a large fraction of events leading to endosomal maturations. Furthermore, our simulations indicate that this emergent mechanism imparts tight temporal control over the ensemble maturation of VEEs.

## Results

### Measuring and quantifying whole cell-level endosomal maturation

To both measure ensemble endosomal conversion dynamics, as well as follow individual endosomes at whole cell level with fast spatio-temporal resolution, we used lattice light-sheet microscopy (LLSM) to image cells expressing APPL1-EGFP[23] and TagRFP-T EEA1[24]. LLSM-based live-cell imaging enabled near-diffraction limited prolonged imaging of ~30 min with a temporal resolution of ~3 s per entire volume of the cell with minimal photobleaching (Fig. 1a; Supplementary Fig. 1a, b; and Supplementary Movie 1). Rapid LLSM imaging confirmed minimal overlap between APPL1 and EEA1 signals with the exception of rapid switch-like APPL1 to EEA1 conversions, as has been reported previously[6]. Visual inspection of the data revealed three major categories of dynamic phenomenology: inter-endosomal 'kiss-and-run' events preceding conversions, inter-endosomal collisions leading to fusion, and conversions (Fig. 1b and Supplementary Movies 1 and 2).

The number of distinct events, and their highly stochastic nature, preclude interpretations based on human-biased selection of representative trajectories. We, therefore, developed an automated image analysis pipeline to convert raw data to full trajectories of detected endosomes, automatically annotated for the presence of events such as heterotypic collisions, conversions, and fusions. Briefly, we identified all potential endosomes using a blob detection routine (Laplacian of Gaussian operator), then filtered to the true endosomes with an unsupervised pattern recognition-based routine (Supplementary Fig. 1c). The brightest and dimmest objects (>100 total) were taken to represent true versus false endosomes, respectively, then used as inputs for template matching to construct a set of features for each class, followed by k-means clustering into signal versus background (Supplementary Fig. 2). These discrete segmented objects were then tracked using a custom tracking routine built with trackpy,[25] using both localisation and intensity values (Supplementary Fig. 1d, Supplementary Movie 2). Tracked objects from opposite channels were then analysed independently to identify collision, fusion, and conversion events based on the time course of spatial separation between nearby endosomes (Supplementary Fig. 1e, f).

### Inter-endosomal interactions are necessary for robust conversions

To investigate whether heterotypic interactions play a regulatory role in very early endosomal maturation, we applied this analysis pipeline to six untreated and two nocodazole-treated whole cell volumes (equivalent to >1 h of total observations), which resulted in the detection of thousands of events. A representative montage of a conversion preceded by multiple collision events is shown in Fig. 1c, with the corresponding intensity trace (with annotated events) in Fig. 1d; note clear jumps in the intensity of EEA1 following collisions at

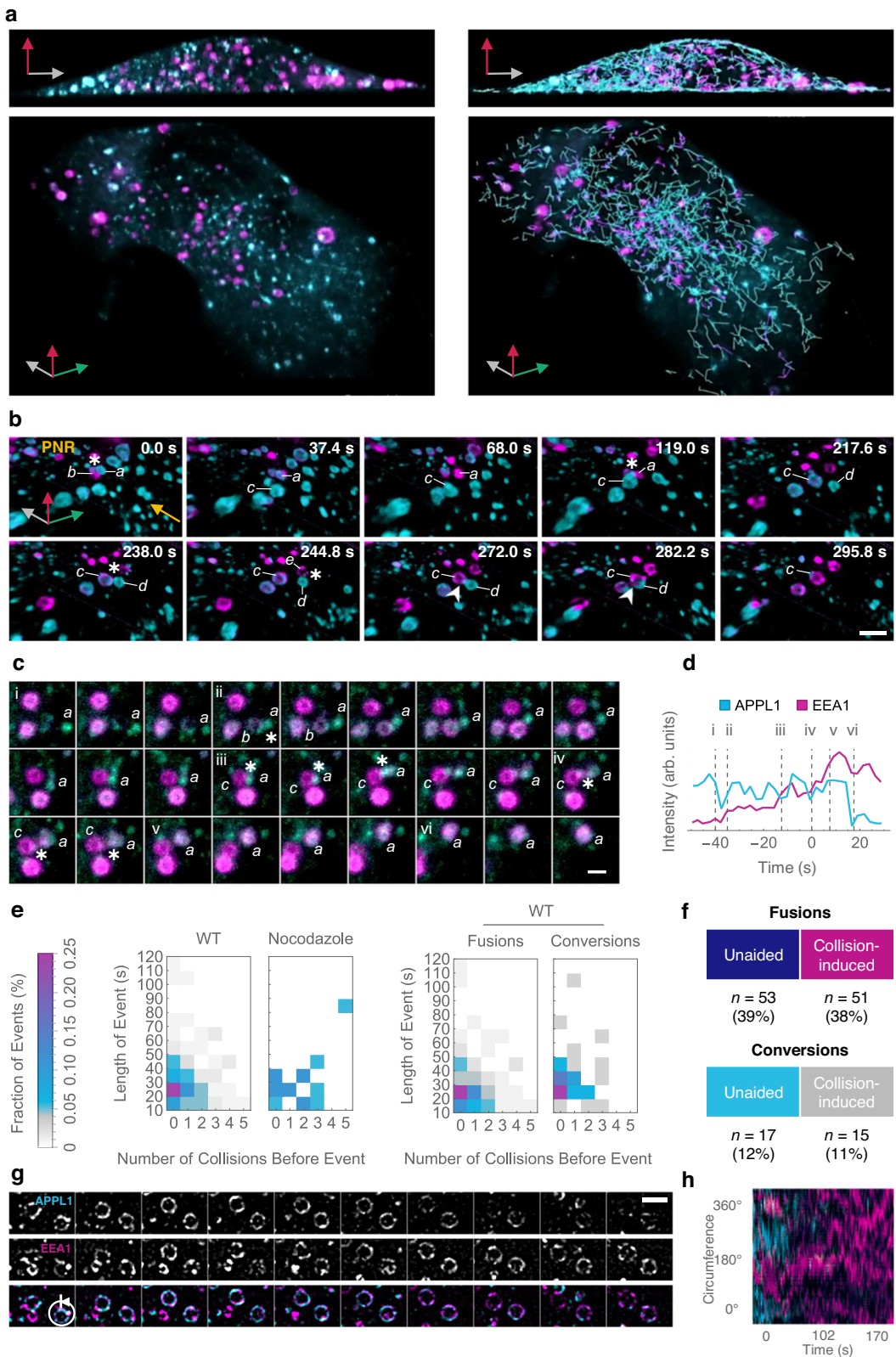

(ii), (iii), and (iv). We applied stringent selection criteria to all automatically identified events to select only clear cases of APPL1 to EEA1 conversion (Supplementary Fig. 3, Supplementary Movie 3), confirmed by visualisation of population-average signals of APPL1 and EEA1 immediately before, during, and after each conversion event. During the process of conversion, APPL1 and EEA1 signals for the same endosome showed average cotracking for ~30 s (Supplementary

Fig. 4a), but with considerable variability. This is demonstrated by separating all events into cohorts defined by the total duration of APPL1–EEA1 cotracking (in bins of 10 s); population averages for each cohort are shown in Supplementary Fig. 4b. During visual inspection of these data, we noticed a clear association between the speed of individual APPL1 to EEA1 conversions and the number of preceding heterotypic collisions. To confirm this observation, we calculated the

**Fig. 1 | High temporal- and spatial-resolution data reveal that heterotypic APPL1–EEA1 interactions promote early endosomal maturation. a** Whole-cell LLSM images of APPL1 (*cyan*) and EEA1 (*magenta*) showing raw data (*left*) and trajectory overlays (*right*). See Supplementary Movie 2 for the dynamic version. **b** Montage showing examples of heterotypic collision and fusion leading to conversion. Letters denote distinct endosomes, asterisks collisions, and arrowheads fusions. Arrow indicates the flow of new endosomes, with more mature endosomes at the perinuclear region (PNR). Endosomes are labelled until the last event of interest. At 0.0 s, APPL1 endosome '*a*' collides with EEA1 endosome '*b*', triggering the conversion of '*a*'. Thereafter, '*b*' exits; conversion of '*a*' completes by 68.0 s. '*a*' collides with APPL1 endosome '*c*' at 119.0 s, which begins to convert. Nascent APPL1 endosome '*d*' undergoes multiple collisions, first with converting endosome '*c*' at 238.0 s then with EEA1 endosome '*e*' at 244.8 s. '*d*' fuses with '*c*' starting at 272.0 s, resulting in a larger fused endosome at 282.2 s; APPL1 signal disappears by 295.8 s. Scale bar = 1 μm. **c** Montage of inter-endosomal collisions leading to APPL1–EEA1 conversion. Letters denote participating endosomes, asterisks collisions, and numerals the corresponding frames in (**d**). Scale bar = 1 μm. **d** Intensity traces for endosome '*a*' in (**c**); events indicated by dotted lines. Time is relative to the start of APPL1–EEA1 cotracking; each channel intensity is the total intensity over the APPL1-masked pixels before, and EEA1-masked pixels after, cotracking. **e** Heatmap of the fraction of events defined by duration of cotracking (in 10-s bins) and the number of preceding heterotypic collisions. (*Left*) All events for wild-type (WT) and nocodazole-treated cells. (*Right*) WT data is split into fusions and conversions. WT: $n = 136$ ($n_{fusion} = 104$ and $n_{conversion} = 32$). Nocodazole-treated: $n = 18$ ($n_{fusion} = 18$). **f** Numbers of events in each category (WT). **g** Montage of live SRRF experiment showing dynamic APPL1 (*cyan*) and EEA1 (*magenta*) clustering. Scale bar = 1 μm. **h** Kymograph of normalised APPL1 (*cyan*) and EEA1 (*magenta*) intensity of converting APPL1 endosome imaged with SRRF, after processing, showing circumference position (°) over time (s).

number of collisions occurring between each APPL1 endosome and any EEA1 endosomes (Supplementary Fig. 5) in the 30 s immediately prior to a detected conversion or fusion event, then segmented the distribution of events from each cotracking cohort according to the number of preceding heterotypic collisions (Fig. 1e, Supplementary Fig. 6). Importantly, all slow detected APPL1 to EEA1 conversions and fusions had few or no potential collisions prior to conversion. The relative numbers of each type of event (collision-induced or unaided, fusion or conversion) are summarised in Fig. 1f. In line with previous models of EEA1-mediated fusion[26,27], 39% of the events displayed immediate fusion following collisions ('unaided fusions'). This could be attributed to EEA1-mediated fusion where EEA1 molecules can bridge two endosomes at the instant of collision, as has been postulated previously[27–30]. 38% of events involved fusions, but that was preceded by at least one heterotypic collision ('collision-induced fusions'). While 12% of events represented unaided conversions, which have been reported earlier and result from direct binding of EEA1 from the cytoplasm, collisions leading to conversions were found to be 11% of all events. Together, these events form the endosomal maturation process. Note that heterotypic fusions result in endosomes with both APPL1 and EEA1 and represent an intermediate step in conversion (vide infra). The quantitative analysis also revealed that unaided conversions were more prominent for larger endosomes (Supplementary Fig. 4c), whereas the heterotypic collisions were a feature of a much broader and smaller size of endosomes that showed stochastic directed runs and transitions to periods of little movement, as has been reported for early endosomes[31]. These results underline the necessity of rapid volumetric imaging and bespoke analysis routines to capture the described processes.

## APPL1 and EEA1 are counter-clustered during conversion

Furthermore, we observed that in nocodazole-treated cells, some endosomes showed vacillating 'back-and-forth' fluctuations between the signals of APPL1 and EEA1, never fully committing to a complete conversion into an EEA1-positive endosome that did not revert (Supplementary Movie 4). Interestingly, a few endosomes displaying EEA1 fluctuations were also 'pulsatile', suggesting the existence of clustering (Supplementary Fig. 7), non-linearity, and binding–unbinding events that corresponded to more than a few molecules. Many endosomal markers and associated proteins including dynein have been reported to exist as clusters on the endosomal surface[32–35]. In addition, phosphoinositide lipids display clustering induced by the binding of specific proteins[36]. Therefore, we reasoned that, given the observed dynamics, APPL1 and EEA1 may display some level of clustering.

To confirm the existence of clusters of EEA1, we performed single-molecule localisation microscopy using EEA1 Dendra-2[37]. We found that EEA1 was not uniformly distributed over the entire surface of the endosomes, but instead formed distinct domains (Supplementary Fig. 8). To confirm this observation in live cells and to investigate the

distribution of EEA1 with respect to APPL1, we performed multi-colour live super-resolution microscopy via super-resolution by radial fluctuations (SRRF)[38] of APPL1 and EEA1 (Supplementary Movie 5). Interestingly, we observed that APPL1 and EEA1 are counter-clustered (Fig. 1g, Supplementary Fig. 9a). Both APPL1 and EEA1 show dynamic localisation with time, but this counter-clustering is maintained through the process of conversion until the APPL1 signal is lost (Fig. 1h, Supplementary Fig. 9b–d).

## Two distinct populations of EEA1 endosomes bound via N- and C-termini exist

Taken together, our experimental observations suggested that heterotypic interactions contribute to the initiation of conversion processes. Therefore, we hypothesised that the inter-endosomal binding ability of the EEA1 homodimer and the presence of heterotypic collisions may work together to seed conversions. EEA1 projects out into the cytoplasm due to its ~200 nm-long coiled-coil domain[27,30]; furthermore, it can bind to endosomal membranes at both its N- and C-terminal ends[39]. Whilst at the C-terminal domain, EEA1 binds to membranes through the coincidence detection of Rab5 and PI(3)P[13,30,40], at the N-terminus, EEA1 solely binds to Rab5 through a zinc finger-binding domain[30,41]. We, therefore, rationalised that in a heterotypic collision, the incident APPL1 endosome would have little to no PI(3)P, and as such the only EEA1 binding that is probable is through N-terminal binding, thus producing encoded precedence in EEA1 N- versus C-terminal binding.

To determine which terminus of EEA1 is bound to the already EEA1-positive endosome, and which domain binds to the incoming nascent endosome, we utilised fluorescence lifetime microscopy (FLIM). We reasoned that N-terminally tagged EGFP-EEA1 combined with an RFP FRET partner could distinguish N- from C-terminal binding using the lifetime of EGFP, since EEA1 is 200 nm in length in its straight conformation, and it binds directly to Rab5 via its N-terminus (Fig. 2a). Multi-scale molecular dynamics simulations also suggest that the coiled-coil domain can extend with a tilt up to 50° from the normal to the endosomal membrane surface when bound using the C-terminal FYVE binding domain[42]. Thus, N-terminal binding will result in decreased fluorescence lifetime due to FRET with Rab5 labelled with RFP, whereas C-terminal binding will show the EGFP lifetime since no FRET will take place. We first investigated whether different populations of EEA1-positive endosomes, bound via N- or C-termini, exist in fixed cells. We found that EEA1 endosomes showed two strikingly distinct populations: C-terminally bound EEA1 that localised closer to the nucleus of the cell (Fig. 2c), and N-terminally bound EEA1 that was predominantly peripherally localised (Fig. 2b, c).

Additionally, we were able to detect these same two populations of endosomes using the inverse FRET pair using Rab5 EGFP lifetime in cells transfected with EEA1-TagRFP, in contrast to cells transfected with only EEA1-EGFP, which showed only a single longer lifetime

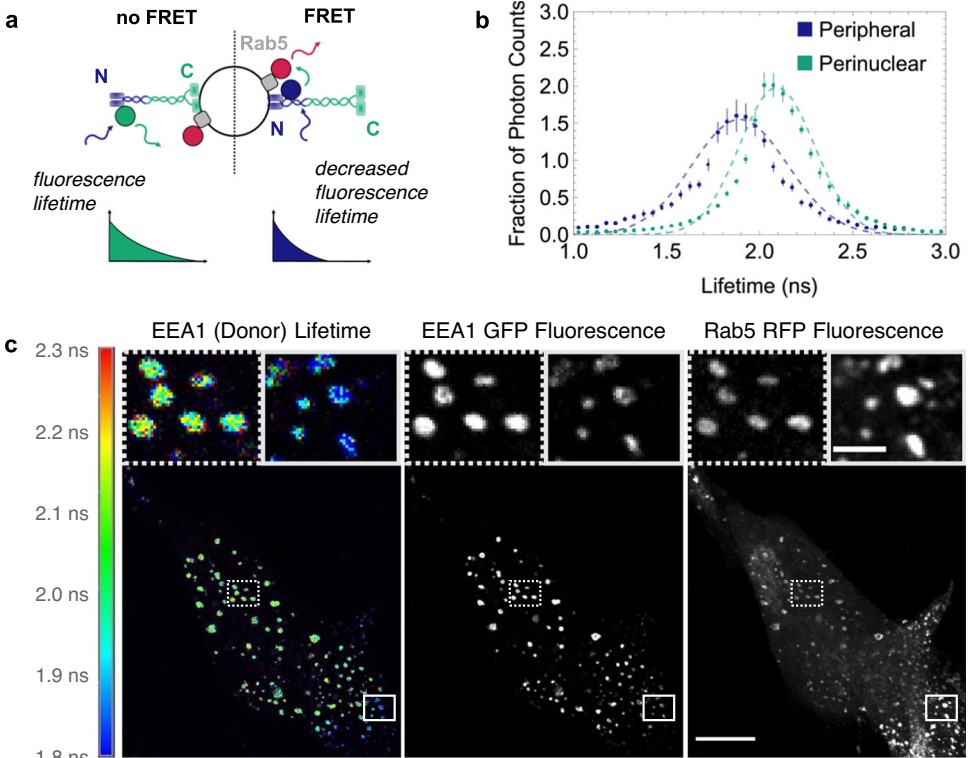

**Fig. 2 | EEA1 endosomes exist in two spatially distinct populations characterised by opposite membrane binding. a** Schematic diagram of EEA1 FLIM experiment logic. A shorter fluorescence lifetime (*right*) indicates a FRET interaction between EEA1-EGFP and Rab5-RFP and therefore indicates EEA1 is bound via its N-terminal binding domain. Correspondingly, a longer fluorescence lifetime (*left*) indicates no FRET interaction and thus that EEA1 is bound via its C-terminal binding domains to the membrane. **b** Normalised frequency histograms of the detected fluorescence lifetimes of EEA1-EGFP photons measured in peripheral endosomes (*blue*) and perinuclear endosomes (*green*); bars represent standard errors of the mean; dashed curves show Gaussian fits for reference. Mean lifetimes calculated from these data are 1.87 and 2.10 ns for the peripheral and perinuclear curves, respectively. Endosomes were measured across *n* = 6 cells. **c** Representative FLIM-FRET experiments of RPE1 cells transfected with EEA1-EGFP and Rab5-RFP. Coloured scale bar represents the donor lifetime ranging from 1.8 ns (*blue*) to 2.3 ns (*red*). Left panel shows the FLIM image of EEA1 (donor) lifetime, the middle panel shows EEA1 fluorescence intensity and the right panel shows Rab5 fluorescence intensity; boxes indicate the regions of the zoomed inset, with dotted lines corresponding to perinuclear endosomes and solid lines to peripheral endosomes. Scale bar = 10 μm. Zoomed insert scale bar = 1 μm.

distribution corresponding to native EGFP (Supplementary Fig. 10). To confirm that these two populations of lifetimes corresponded to N- and C-terminally bound EEA1, we expressed Rab5 EGFP and either CT-Mut EEA1 TagRFP or NT-Mut EEA1 TagRFP, which both showed only a single lifetime peak corresponding to entirely FRET or non-FRET lifetimes, respectively (Supplementary Fig. 10). In addition to the N- and C-terminal mutants, the donor:acceptor intensity ratio versus fluorescence lifetimes displayed no dependence, confirming that the observed lifetime decrease results from FRET interactions and not insufficient acceptor molecules (Supplementary Fig. 11). These experiments strongly indicate that, in newly generated endosomes, the first EEA1 binding occurs via the N-terminus.

### EEA1 binding via the N-terminus precedes binding via the C-terminus

To map the temporal dynamics of EEA1 binding via the N- or C-termini, we performed live-cell FLIM of EGFP-EEA1 and Rab5-mRFP. However, live-cell FLIM using confocal microscopy with a sufficient temporal resolution to capture endosomal processes intrinsically results in a reduced number of collected photons. To overcome this, we took advantage of a priori knowledge from fixed cell experiments and fit live FLIM data with the two-lifetime components detected in fixed experiments. This gave a shorter lifetime component corresponding to N-terminally bound EEA1, where GFP can 'FRET' with Rab5-RFP, and a longer fluorescence lifetime corresponding to C-terminally bound EEA1, where the N-terminus is at least 150 nm away, extended into the

cytoplasm from the Rab5 RFP. We then separated the detected photons collected at each pixel based on these two components, effectively giving an 'NT EEA1' and a 'CT EEA1' channel (Fig. 2a). Using this two-component fitting, we visualised the initial appearance of EEA1 on Rab5-positive, EEA1-negative endosomes following a collision–conversion event. We observed that only N-terminally bound EEA1 (Fig. 3a,b; Supplementary Movies 6 and 7) localised on these Rab5-positive endosomes and displayed an increasing signal of C-terminally bound EEA1, concomitant with fusions and trafficking towards the perinuclear region (PNR) (Fig. 3d, e; Supplementary Movie 8). This gradual acquisition of C-terminal EEA1 seen through the increase in longer lifetime components and reduced N:C intensity ratio (Fig. 3e) suggests a concurrent phosphoinositide conversion of PI(3,4)P2 into PI(3)P, with the initial trigger via N-terminally bound EEA1, even for unaided conversions. This subsequent maturation following the appearance of N-terminal EEA1 can also be observed with analogous FLIM analysis methods including phasor plots and average pixel lifetimes (Supplementary Fig. 12).

Whilst EEA1-EEA1 fusions are commonly observed, by separating EEA1 vesicles into their constituent N- and C-terminally bound populations, we observed that fusions primarily occurred when at least one vesicle had C-terminal EEA1 present (Fig. 3c). Fusions were most likely to occur between N- and C-terminal EEA1-positive or C- and C-terminal EEA1-positive endosomes (Fig. 3c). Endosome pairs with at least one EEA1-negative endosome did not show significant fusions. Remarkably, in cases with both endosomes N-terminally

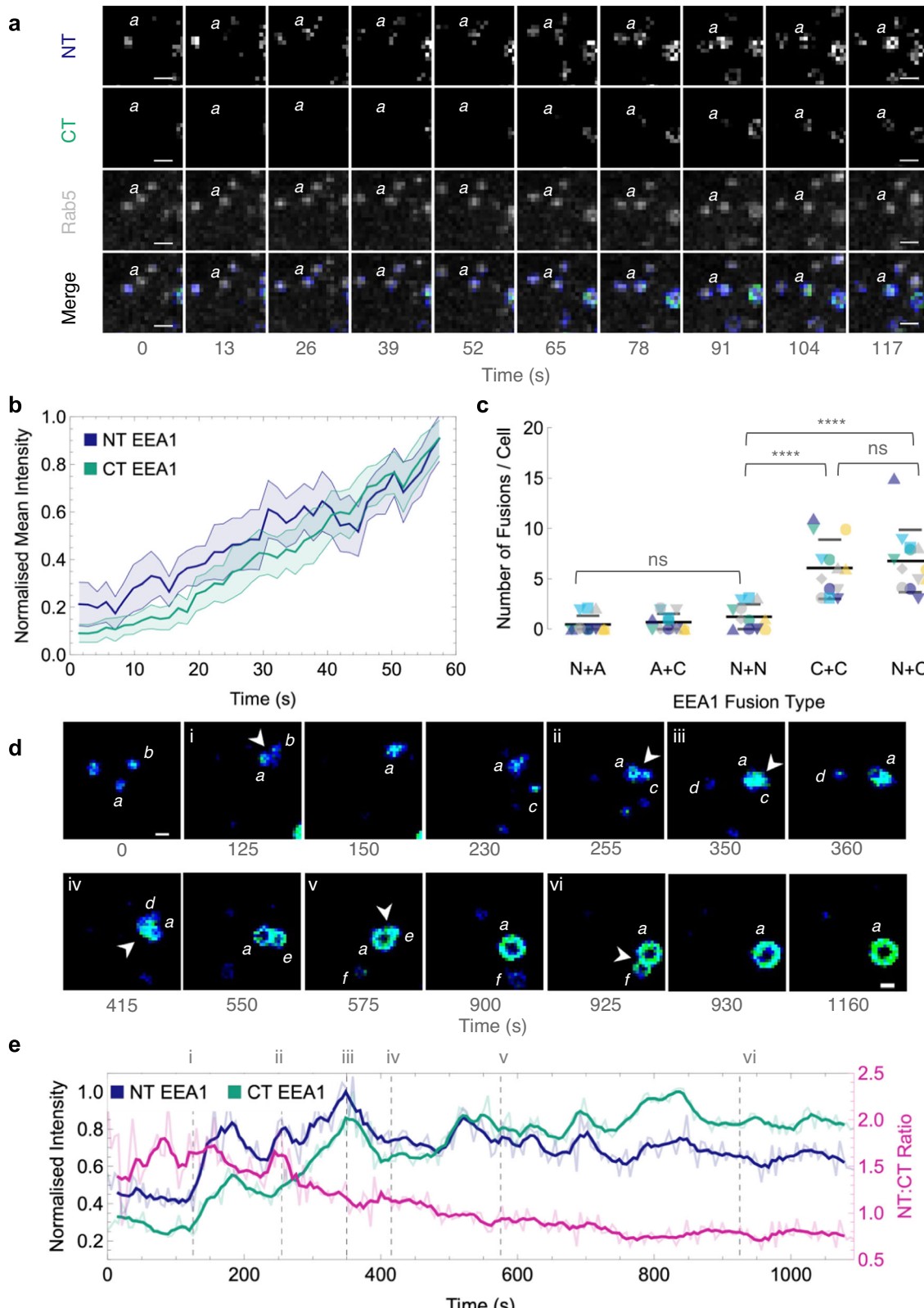

positive, no significant fusions were observed. Three conclusions could be drawn from these results. Firstly, the requirement of at least one C-terminally bound EEA1 and non-fusion of N-terminally bound EEA1-positive endosomes suggests that cross-binding of EEA1 is a necessary step for endosomal fusions to occur. This aligns with previously published results showing that both ends of the endosomal tether must be stably bound to result in endosomal fusion[27,30].

Secondly, the appearance of N-terminally bound EEA1 prior to C-terminally bound EEA1 in cases of unaided conversions indicates that the N-terminal binding is a necessary and intermediate step before further undergoing a maturation via phosphoinositide conversion into C-terminally bound EEA1. Finally, with the requirement of at least one EEA1-positive endosome being C-terminally bound, and the other being either N- or C-terminally bound, the

**Fig. 3 | EEA1 initially binds via its N-terminal binding domain before maturing into a C-terminal bound endosome. a** Representative montage showing the appearance of N-terminal EEA1 following collision-conversion, imaged by live FLIM-FRET as described in Fig. 2a. EEA1-EGFP fluorescence lifetime was fit with a two-component regression ($\tau_1 = 1.006$ ns, $\tau_2 = 2.600$ ns). The image is pseudo-coloured by the relative photon contribution from each component; the shorter N-terminal EEA1 component (*blue*), the longer C-terminal EEA1 component (*green*), and Rab5-RFP fluorescence (*grey*). '*a*' indicates nascent converting endosome. Scale bar = 2.5 μm; time is measured in seconds. **b** Normalised mean intensity plot of converting endosomes following seeding at time = 0. Line graphs show the mean intensity of N-terminally (*blue*) and C-terminally (*green*) bound EEA1, area fill indicates a 95% confidence interval at each time-point. Time is measured in seconds. *n* = 20 conversion events. **c** Fusions categorised by the participating endosomes: N-terminal bound (N); C-terminal bound (C); and EEA1 absent, Rab5-positive (A). Plots show mean, error bars indicate SD. Each coloured shape indicates a different cell, *n* = 13. ns indicates non-significant difference, **** indicates *p* < 0.0001. Each

mean was compared against the others using an ordinary one-way ANOVA with Tukey's multiple comparisons. N + A vs. A + C *p* = 0.9984, N + A vs N + N *p* = 0.8678, N + A vs. N + C *p* = $5.35 \times 10^{-8}$, N + A vs. C + C *p* = $1.17 \times 10^{-6}$, A + C vs. N + N *p* = 0.9603, A + C vs. N + C *p* = $1.55 \times 10^{-7}$, A + C vs. C + C *p* = $3.06 \times 10^{-6}$, N + N vs. N + C *p* = $1.62 \times 10^{-6}$, N + N vs. C + C *p* = $2.52 \times 10^{-5}$, N + C vs. C + C *p* = 0.9058. **d** Representative montage of EEA1 N- to C-terminally bound conversion, showing N-terminally (blue) and C-terminally (green) bound EEA1. Letters denote distinct endosomes, with endosome '*a*' corresponding to the intensity trace in (**e**); arrowheads indicate fusion events, enumerated as in (**e**). Scale bar = 2.5 μm; time is measured in seconds. **e** Intensity trace of N- to C-terminally bound conversion corresponding to montage in (**d**). Lines indicate mean relative lifetime amplitudes of N-terminal EEA1 (*blue*), C-terminal EEA1 (*green*), and the N:C intensity ratio (*magenta*); bold lines indicate 3-frame moving average; intensities were manually measured for each endosomal pixel. Dotted lines indicate successive fusion time points, enumerated as in (**d**). Time is measured in seconds.

EEA1-mediated fusion of endosomes is biased towards the more mature, later populations.

## Endosomal conversions are driven by phosphoinositide conversions by INPP4A

To further characterise the maturation into EEA1 endosomes, with N-terminal EEA1 binding preceding C-terminal EEA1 binding in the context of phosphoinositide conversion, we combined the FLIM-based investigation of EEA1 orientation with staining for PI(3)P in fixed cells. To label PI(3)P without inducing overexpression artefacts or steric hindrance, we utilised a purified recombinant GST-2xFYVE probe that could be detected using antibodies against GST as described previously[43,44]. We observed that C-terminally bound EEA1 endosomes have significantly higher PI(3)P labelling as compared to N-terminally bound EEA1 endosomes or the peripheral Rab5-positive, EEA1-negative endosomes (Fig. 4a, Supplementary Fig. 13). This is in agreement with previously published studies of EEA1 C-terminal coincidence detection between Rab5 and PI(3)P[39,45,46], and suggests that NT-EEA1 appearance may precede PI(3)P production on endosomes.

The two distinct modes of EEA1 binding via the N- and C-termini, and the fraction of unaided conversions of APPL1 to EEA1 observed using LLSM, suggested that phosphoinositide conversion that results in PI(3,4)P2 to PI(3)P must occur on the incoming nascent endosomes. This hypothesis is supported by live-FLIM data, which showed that a corresponding fraction of endosomes displayed an N- to C-terminally bound EEA1 exchange, strongly suggesting that the source of PI(3)P must be within the same endosomes that have not collided with more mature endosomes. However, it was unclear whether this PI(3)P production triggered during early endosomal maturation was produced through dephosphorylation of PI(3,4)P2 or phosphorylation of PI. To distinguish these possibilities, we targeted INPP4A, a PI4-phosphatase that dephosphorylates PI(3,4)P2 to PI(3)P, as well as VPS34, a class III PI3-kinase that phosphorylates PI to generate PI(3)P and is another source of PI(3)P at the early endosomal level[47,48]. To test whether PI(3)P generated via VPS34 contributes to APPL1 to EEA1 conversions, we used SAR405, a drug that specifically targets VPS34[44,49]. Quantifying and comparing the number of conversions versus untreated cells revealed that SAR405 treatment caused a 3-fold reduction in the number of detected conversions. In contrast, targeting INPP4A using siRNA caused a more severe ~10-fold reduction in the number of detected conversions, suggesting that most conversions were driven by PI(4,5)P2 to PI(3)P conversion via INPP4A (Fig. 4b). Despite the distinct effects on early endosomal maturation rate of INPP4A siRNA and SAR405, these treatments led to a similar 50–60% reduction in Rab5-localised PI(3)P, highlighting that PI(3)P produced via PI3-kinase or PI4-phosphatases play complementary roles in early endosomal biology (Supplementary Fig. 14). Consistent with these results, upon assaying for the binding of EEA1 using FLIM to distinguish between

N- versus C-terminal binding, we found a clear reduction in the number of C-terminally bound EEA1-positive endosomes in SAR405 treated cells, but never a complete abolishment, suggesting that INPP4A-mediated phosphoinositide conversions acted as a source for a fraction of PI(3)P on these early endosomes (Supplementary Fig. 15).

In addition to the impact on endosomal maturation, inhibition of PI(3)P production by VPS34 or INPP4A using SAR405 or siRNA leads to a significant reduction in early endosomal fusion rates, highlighting the central role EEA1 dual-endosome binding plays in this process (Fig. 4b). It is interesting to note that whilst the SAR405-treated cells displayed almost no fusion events, consistent with a drastic loss of PI(3)P and therefore impaired EEA C-terminal binding, INPP4A siRNA-treated cells retained ~20% of their fusions, suggesting that a population of transiently N-terminally bound EEA1 vesicles were still able to fuse with more mature early endosomes containing VPS34-produced PI(3)P.

## N-terminal binding of EEA1 is necessary for endosomal maturation

To validate the consistent observation of N-terminal binding of EEA1 prior to any maturation process, and to investigate the stringency of the requirement for N-terminal binding of EEA1 via Rab5 in maturation, we used an N-terminal mutant of EEA1 carrying F41A and I42A at the C2H2 $Zn^{2+}$ site (EEA1 Nt-mut), which is impaired in Rab5 binding[14] (Fig. 4c). When expressed in wild-type RPE1 cells, conversions were unimpaired and endosomal fusions were only mildly affected. This suggested that the Rab5 binding mutant, EEA1 Nt-mut, did not display a strong dominant negative phenotype and that the endogenous EEA1 could still function to evince endosomal maturations (Fig. 4d). This could be because the observed clustering buffers against dysfunctional mutant EEA1; in addition, as EEA1 is a homodimer, it may still have one active binding site. Therefore, we used a HeLa EEA1 knockout (KO) cell line and transiently expressed EEA1 Nt-mut. In contrast to wild-type EEA1, EEA1 Nt-mut exhibited no heterotypic interactions resulting in maturations per cell over 20 min. Furthermore, no EEA1 signals were observed on APPL1 endosomal trajectories, suggesting that the collision-triggered conversion mechanism was dysfunctional owing to impaired Rab5 binding at the instance of collision. It is also to be noted that the expression of EEA1 Nt-mut resulted in larger but fewer and less motile endosomes (Supplementary Movie 8). If only the C-terminus of EEA1, via its Rab5 and FYVE binding, were involved in the phosphoinositide-governed conversion, we would expect to detect some number of APPL1 to EEA1 conversions. In our experiments, unaided conversions were also completely abrogated, indicating that, even in direct conversions, where collisions may not play a role, the N-terminal binding is a compulsory intermediate step.

The C-terminus of EEA1 harbours a FYVE domain and a Rab5 binding domain. Unfortunately, our attempts to investigate the role of

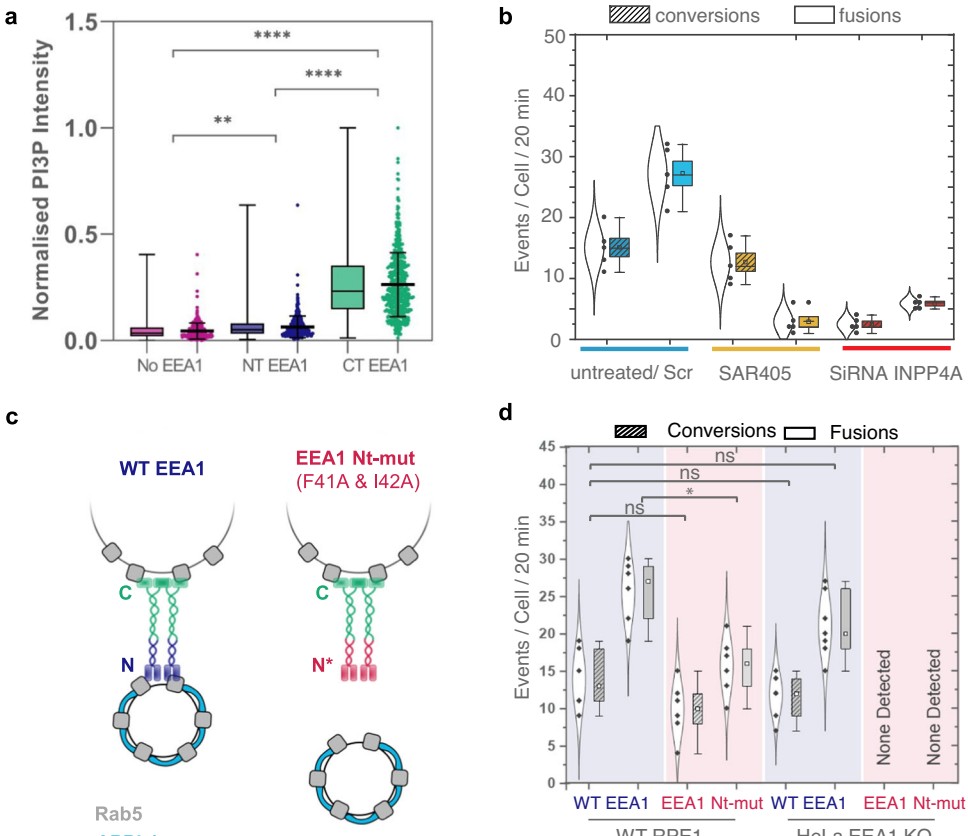

**Fig. 4 | The initial N-terminal of EEA1 is essential for endosomal conversions.** **a** FLIM of orientation with phosphoinositide lipid specificity. Scatter and box-and-whisker plots of PI(3)P staining intensity by endosome type; Rab5 positive and EEA1 negative (*magenta*), NT EEA1 bound (*blue*), or CT EEA1 bound (*green*). PI(3)P intensities are measured using 2xFYVE-GST labelling and normalised across each cell. Error bars on scatter plots indicate mean ± S.D. Statistical significance determined using one-way ANOVA, ** indicates $p < 0.01$, **** indicates $p < 0.0001$ (EEA1 vs. NT EEA1 $p = 0.0086$, No EEA1 vs. CT EEA1 $p = 1.32 \times 10^{-6}$, NT EEA1 vs. CT EEA1 $p = 3.45 \times 10^{-5}$) $n = 18$ cells. Single points indicate measured data, and the box plots correspond to the 25th to 75th percentile of events, with bars showing the total range. **b** SAR405-treated cells show reduced fusions between APPL1 and EEA1 endosomes, but not reduced APPL1 to EEA1 conversions. siRNA-INPP4A-treated cells show reduced fusions and conversions. Single points indicate measured data; the violin plots correspond to a normal distribution of all events; and the box plots correspond to the standard error of the mean, with bars showing the total range. Means are depicted by the open squares. Statistical significance was determined using one-way ANOVA (Conversions: untreated vs. SAR 405. $p = 0.2936$, Conversions: untreated vs SiRNA INPP4A. $p = 1.45 \times 10^{-5}$, Fusions: untreated vs SAR405 $p = 0.36 \times 10^{-5}$, Fusions: untreated vs SiRNA INPP4A. $p = 0.85 \times 10^{-5}$; $n = 5$ cells per condition. **c** RPE1 wild-type cells and HeLa EEA1 knockout (KO) cell lines expressing wild-type EEA1 (*blue*) or N-terminal mutant deficient in binding Rab5 (*red*) were imaged using LLSM. **d** The total number of conversions and fusions were quantified for WT EEA1 (blue shade) and EEA1 Nt-mutant (Grey shade); these data indicate that the initial N-terminal of EEA1 is essential for endosomal conversions. ns indicates non-significant difference, * indicates $p < 0.05$; $p = 0.0018$. Each mean was compared against the others using an ordinary one-way ANOVA. In the case of HeLa EEA1 KO cells expressing EEA1 N-terminal Rab5 binding mutant, no events were detected by the analysis workflow or by visual inspection. Single points indicate measured data; the violin plots correspond to a normal distribution of all events; and the box plots correspond to the standard deviations of events, with bars showing the total range. Means are depicted by the open squares ($n = 6$ cells per condition).

PI(3)P binding in conversions using a construct with a mutation in the C-terminal PI(3)P binding pocket (R1375A)[46] proved unfruitful. We observed that the localisation of this mutant was largely cytosolic with quick transient binding in some cases, as has been reported elsewhere[46]. This prevents any direct measurement of the influence of FYVE domain-based PI(3)P binding on the entire process of conversion. However, it emphasises the role of PI(3)P binding by the FYVE domain, along with Rab5, in localising EEA1 robustly to the endosomes, in agreement with previously suggested models of dual interactions/coincidence detection of the EEA1 C-terminus[7,39,46].

### A feed-forward endosomal conversion model

To summarise, collisions between endosomes form an important step in overall endosomal conversion rates. The live FLIM data suggests that N-terminally bound EEA1, via interaction with Rab5, is a step preceding the phosphoinositide-based binding of EEA1 via its C-terminal FYVE domain (Fig. 3). Expressing the N-terminal Rab5 binding mutant in HeLa EEA1 KO did not rescue any maturation events, suggesting that this is a necessary step (Fig. 4). Additionally, super-resolution imaging suggests clustered distribution of EEA1, as well as counter-clustering of APPL1 to EEA1 (Fig. 1g, h and Supplementary Figs. 5 and 6). This suggests the presence of feedback in the reaction scheme that governs progressively preferential EEA1 binding over APPL1 binding. To construct a plausible model that agrees with our experimental observations as well as the known protein–protein and protein–membrane interactions of the components involved, we designed a computational model that captures the complex interplay between the distinct phosphoinositide molecules, Rab5, APPL1, and EEA1, and the phosphoinositide conversion (Fig. 5). Importantly, we took into consideration the N-terminal domain of EEA1, which was observed to bind first in unaided collisions as well as in aided conversions through collisions. To simulate this system, we used a grid on the surface of a sphere with two layers of nodes, consisting of a layer of Rab5 and a phosphoinositide layer which began as PI(3,4)P2 but could be converted to PI(3)P by INPP4A if unbound[17]. Binding to these layers

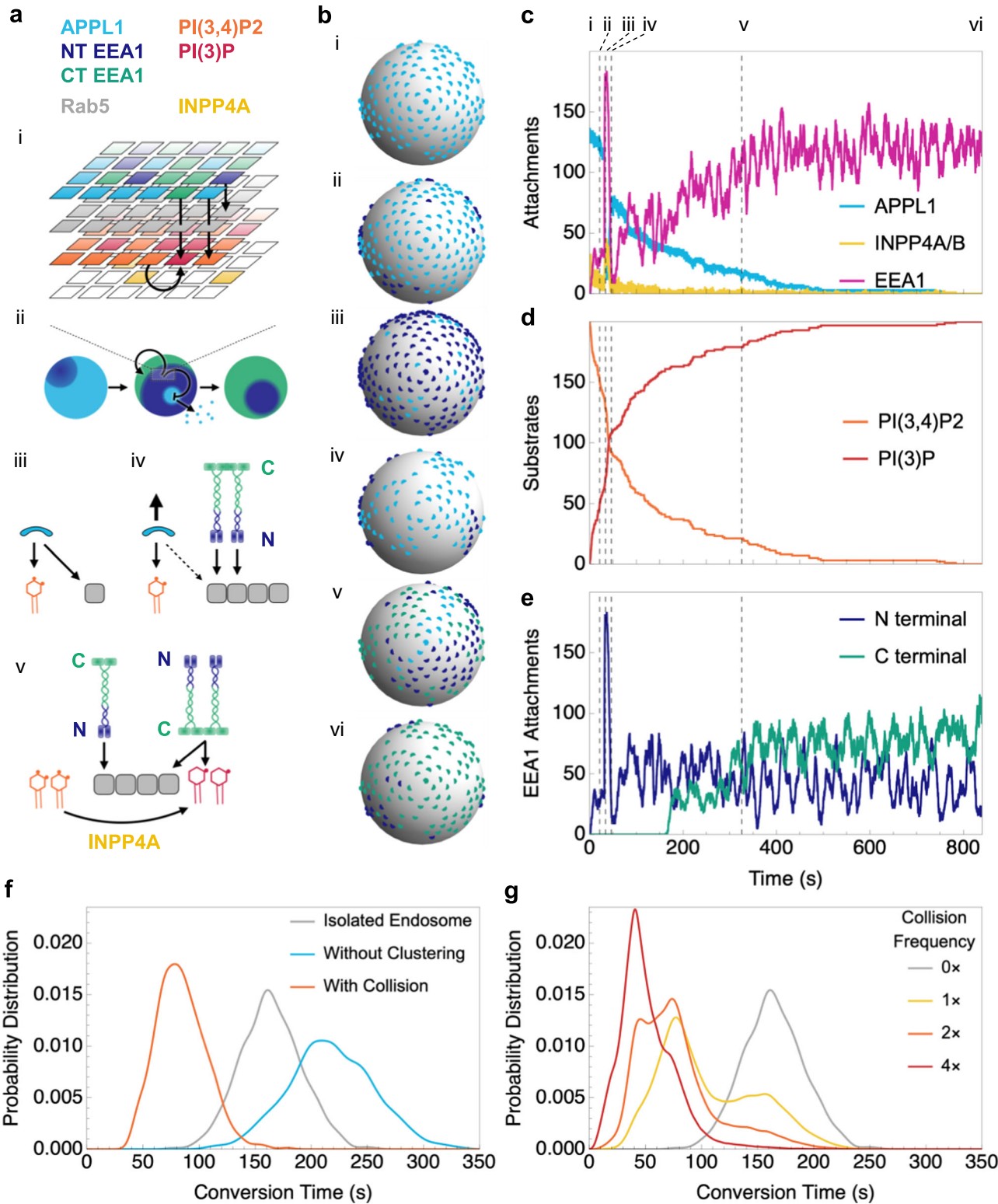

of nodes were the agents, each with a different attachment and detachment rate depending on the nodes present: APPL1 binding Rab5 and PI(3,4)P2; N-terminal EEA1 binding Rab5; and C-terminal EEA1 binding Rab5 and PI(3)P. The interaction map of agents and nodes is shown in Fig. 5a. Using this reaction scheme, we were able to simulate the reactions and tune the parameters to recapitulate the experimentally observed conversion dynamics, as well as formulate the effects of the 'trigger and convert' mechanism (Supplementary Movie 9).

Figure 5b–e shows an example trajectory, beginning with a very early endosome that is APPL1-positive and bound to PI(3,4)P2 and Rab5 via its PH-BAR domain. Spontaneous binding of INPP4A to this endosome can result in the conversion of PI(3,4)P2 to PI(3)P; however, with APPL1 occupying most PI(3,4)P2, most INPP4A remains unbound and, therefore, inactive on its substrate. APPL1 can be transiently displaced by N-terminal EEA1, which binds directly to Rab5. Due to the inclusion of a positive feedback switch to mimic the experimentally observed clustering, APPL1 endosomes are relatively stable (i). However, upon

**Fig. 5 | Agent-based simulations show the effect of clustering and collision on endosome conversion time. a** Schematic of agent and node logic used in modelling. (i) The endosome is simulated as a multi-layered spherical surface with layers occupied by different agents and nodes. The sphere is initially dominated by APPL1 attachments (*cyan*), which are replaced by N-terminal EEA1 (*dark blue*), and finally C-terminal EEA1 (*green*). (ii) APPL1 and EEA1 stochastically bind and unbind, competing for Rab5 (*grey*) binding availability. (iii) APPL1 requires both Rab5 and an adjacent PI(3,4)P2 (*orange*) to attach. (iv) N-terminal EEA1 replaces APPL1 in binding to Rab5 and frees up PI(3,4)P2. (v) INPP4A (*yellow*) converts free PI(3,4)P2 to PI(3)P (*red*). C-terminal EEA1 requires Rab5 and adjacent PI(3)P to bind. **b** Example maturation time course of simulated endosome undergoing a collision, showing attached APPL1 (*cyan*), N-terminal EEA1 (*dark blue*), and C-terminal EEA1 (*green*) at different time-points (dashed lines in **c**–**e**). Initially, the endosome is dominated by APPL1 (i), while N-terminal EEA1 attaches in clusters (ii). Upon collision ($t = 31.5$ s), there is a large influx of EEA1 N-terminal attachments (iii) followed shortly by cluster

detachment (iv). This duration suffices to displace a large number of APPL1, allowing INPP4A to accelerate the conversion of PI(3,4)P2 to PI(3)P. Eventually, C-terminal EEA1 attaches in clusters (v) and dominates over N-terminal attachments (vi). **c**–**e** Time series of numbers of **c** attached APPL1, INPP4A, and EEA1; **d** PI(3,4)P2 and PI(3)P; and **e** N- and C-terminally attached EEA1. See Supplementary Movie 9 for the dynamic version. **f** Conversion time distributions of simulated endosomes in isolation (*grey*), without clustering (*cyan*), and with collision (*orange*). **g** Conversion time distributions of simulated endosomes colliding randomly at the indicated collision frequencies; increased collision frequencies lead to faster conversions. There are multiple modes in the conversion time distributions, corresponding to the number of collisions the endosome experienced before conversion. The leftmost mode at 20–50 s corresponds to two collisions before conversion, the middle mode at 60–100 s to a single collision, and the rightmost mode at 130–200 s to zero collisions.

the introduction of a large pool of EEA1 as the result of a collision (ii), N-terminal EEA1 can sequester Rab5, thus destabilising the APPL1-Rab5 interactions and resulting in APPL1 desorption (iii). Consequently, INPP4A can now bind to its substrate PI(3,4)P2 and convert it to PI(3)P (iv). This leads to the binding of EEA1 through its C-terminal FYVE binding domain, as well as Rab5 binding (v, vi). In this scheme, the N-terminal binding of EEA1 acts as a trigger. Moreover, since the N-terminus of EEA1 has a weak binding affinity to Rab5, we reasoned that the clustered organisation of EEA1 on endosomes, and the interaction of multiple N-terminal EEA1 at the instance of collision, would result in overwhelming the APPL1–Rab5 on the incoming endosome. We simulated the net decrease in conversion time of a single endosome that underwent one collision (Fig. 5f), and the net decrease in conversion time of endosomes in a cell allowed to collide randomly at increasing collision frequencies (Fig. 5g).

These agent-based simulations showed that clustering has a two-pronged effect on accelerating conversion. If a Rab5 molecule originally surrounded by bound APPL1 is occupied by EEA1, it will become unavailable for binding to APPL1. This creates a 'hole' in the APPL1 layer, which decreases the binding affinity of APPL1 in the region surrounding the hole (as compared to a region filled by APPL1 since clustering increases the binding affinity of a species in accordance with the local density of that species). This in turn increases the chance that the hole will expand. On the other hand, clustering of EEA1 attracts more EEA1 to the vicinity of the 'hole'. These two factors speed up the local back-and-forth conversion between APPL1 and EEA1 clusters, which increases the windows of opportunity for INPP4A to convert PI(3,4)P2 to PI(3)P. A heterotypic fusion between endosomes with N-terminally bound EEA1 and C-terminally bound EEA1, as observed in the live FLIM experiments (Fig. 3a) represents only a state with a higher N- to C-EEA1 ratio and the reaction scheme will proceed to convert the transiently increased PI(3,4)P2 to PI(3)P, subsequently replacing N-terminally bound EEA1 by C-terminally bound EEA1. Through our simulations, we were able to quantify the net decrease in conversion time due to clustering (Fig. 5f). Once a sufficient number of PI(3,4)P2 have converted to PI(3)P, C-terminal attachments dominate since they have a stronger binding affinity, and they require both PI(3)P and Rab5 to bind, rendering Rab5 unavailable for N-terminal attachments.

## Discussion

The endosomal system is highly dynamic, requiring successive biochemical maturations of key lipids and associated proteins to achieve correct targeting of internalised cargo. Whilst the order of appearance of key species has been diligently identified for early endosomes, how the timing of maturation is maintained for each generated vesicle had not been studied. In this work, we describe a mechanism that ensures timely maturation of vesicles at a whole cell level. Specifically, we present a trigger-and-convert mechanistic model of APPL1 to EEA1 early endosomal maturation, as summarised in Fig. 6.

In this model, nascent very early endosomes (VEEs) characterised by APPL1 bound to PI(3,4)P2 and Rab5[12] undergo active transport along microtubules and collide stochastically with mature EEA1-positive early endosomes (EEs). This collision is a 'trigger' that primes the VEE for maturation. Our experimental observations are consistent with a model whereby a cluster of EEA1 is transferred onto the incident VEE following such a collision. Furthermore, this model is in accordance with the following molecular details of EEA1. First, C-terminal EEA1 has a rigid quaternary structure that ensures that the coiled-coil region extends into the cytoplasm, preventing the N-terminus from folding back and binding to Rab5 on the same endosome[27]. This would result in the N-terminus of EEA1 being located 160–180 nm from the endosome surface[30], in agreement with observations of two distinct EEA1 populations made in our FLIM experiments. Second, EEA1 possesses two distinct Rab5-binding sites—one corresponding to the C2H2 Zn$^{2+}$ finger at the N-terminus and the other overlapping with the PI(3)P binding FYVE domain at the C-terminus. The C-terminal end also contains a calmodulin (CaM) binding motif. Of EEA1's two Rab5-binding domains, the N-terminus forms a stronger interaction in isolation[14,39]; however, in the presence of PI(3)P, the FYVE domains at the C-terminus of EEA1 lead to a much stronger association with endosomal membranes by coincidence detection of Rab5 and PI(3)P[7,46]. While the exact steps at the instant of collision fall beyond the scope of this manuscript, it is conceivable that a collision would result in the stronger N-terminus–Rab5 interaction overriding the C-terminus interactions. Furthermore, an unexplored but plausible mechanistic detail lies in the interactions of Ca$^{2+}$/CaM with Rab5 and the C-terminus of EEA1, which antagonises PI(3)P binding, and may operate to release C-terminal binding when the N-terminal interactions take place as a result of the collision[50,51]. Whether transient Ca$^{2+}$ spikes operate to mediate the transfer of molecules remains an attractive detail to investigate.

After the collision, the sequestration of Rab5 via N-terminal EEA1 results in the desorption of APPL1 clusters. The reduced APPL1 binding to Rab5 also exposes PI(3,4)P2 to dephosphorylation by 4-phosphatases, producing PI(3)P. The most likely candidate for this reaction is INPP4A since it localises to Rab5-positive EEs[8,17]. This availability of PI(3)P now enables EEA1 to bind via its C-terminal FYVE domains, thereby resulting in the irreversible maturation of an EEA1-positive EE. This mature endosome is in turn able to trigger more conversions of APPL1 VEEs following collisions, thus ensuring continual maturation of this dynamic population of vesicles.

Consistent with other studies of descriptions of specific domains on endosomes, we observed that both VEEs and EEs showed a counter-clustered APPL1 and EEA1 distribution. The hypothesis that clustering plays a key role in ensuring a more robust process was recapitulated through our simulations, which suggested it to be essential for the timely conversion of these vesicles. An attractive hypothesis is that phosphoinositide clustering underlies the observed protein distributions, as phosphoinositide clustering has been demonstrated in other

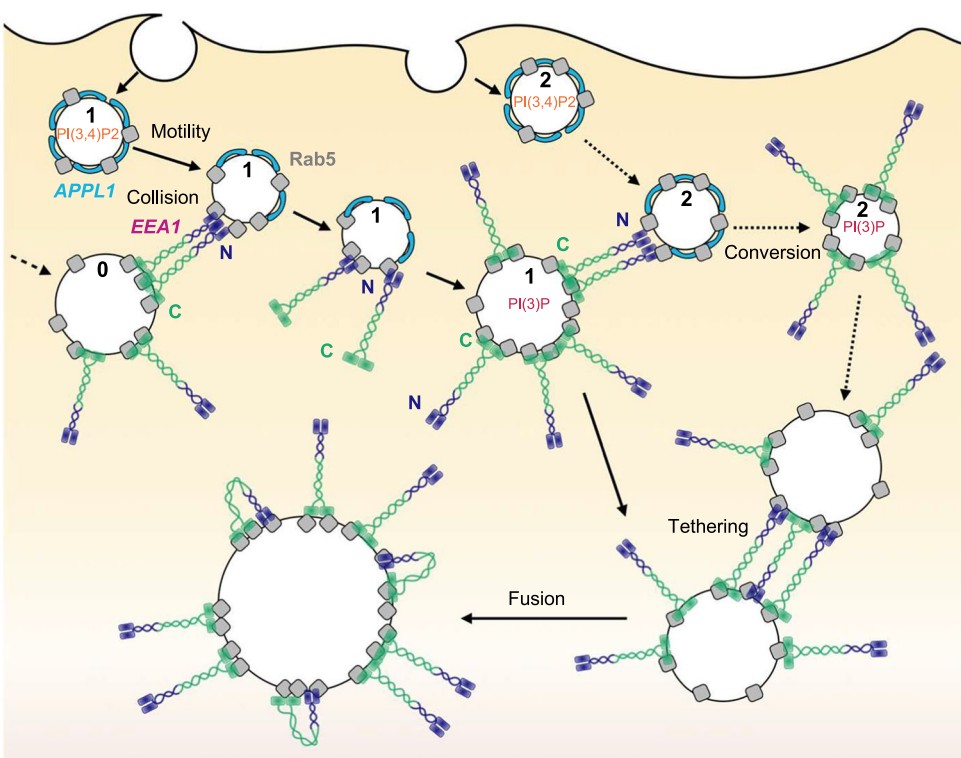

**Fig. 6 | Summary of proposed EEA1 'trigger-and-convert' mechanism of maturation.** Very early endosomes formed at the cell periphery (endosome 1) have PI(3,4)P2 (*orange*)-containing membranes and APPL1 (*cyan*) bound to Rab5 (*grey*). These vesicles collide with mature EEA1 vesicles (endosome 0), seeding N-terminally bound EEA1 and triggering the conversion process. This enables the production of PI(3)P (*red*) and the binding of C-terminal EEA1. These vesicles can trigger conversions on nascent APPL1 vesicles (endosome 2) and participate in canonical endosomal tethering and fusion processes (bottom endosomes).

vesicular and tubular membrane entities[36]. Additionally, Rab5 has also been suggested to be clustered[35]. A clustered distribution of EEA1 or its binding partner Rab5 in the incident endosome would ensure that a higher probability of transfer of EEA1 molecules exists following a collision. Furthermore, this would produce large fluctuations of EEA1 intensity on a converting endosome, as observed in our imaging movies.

Previous studies have shown that stochastic fluctuations have a significant effect on trafficking and maturation processes[19]. The greater the stochasticity in a system, the more the system dynamics favour non-steady state biochemical maturation over steady state vesicular exchange in cellular transport pathways. Biochemical maturation is characterised by a first passage time event in which the first instance of complete maturation of the compartment in question marks a point of no return. But the noise due to the inherent stochasticity in the system poses challenges to the robust directional flow of material, which requires tight regulation of exchange processes between organelles. It was shown by Vagne and Sens that the presence of positive feedback in the maturation process can significantly suppress the stochastic fluctuations, and Golgi cisternae use homotypic fusions as the likely mechanism to overcome this challenge[19]. In a similar vein, our proposed mechanisms of clustering, collision, and heterotypic fusion each provide positive feedback to the maturation process and are essential in the robust functioning of the exchange processes through noise suppression.

The specific requirement for INPP4A, which converts PI(3,4)P2 to PI(3)P on the maturing endosome, ensures a definitive distinction between APPL1- and EEA1-positive endosomes. This is achieved by the depletion of PI(3,4)P2, which ensures that APPL1 cannot rebind following desorption, and thus that conversions are unidirectional. Therefore, even though VPS34-mediated conversion of PI to PI(3)P forms the major source of PI(3)P, we hypothesise that a more significant role is played in the process of APPL1 to EEA1 maturation by virtue of depletion of PI(3,4)P2 and subsequent enrichment of PI(3)P even before the newly generated endosomes have fused with endosomes bearing VPS34-derived PI(3)P. This is consistent with a recent study where treating cells with selective inhibitors of PI3KC2α, Phosphatidyl Inositol Three-kinase Class twO INhibitors (PITCOINs), resulted in a reduction in endosomal PI(3)P, along with reduced EEA1 levels[17,52].

Early endosomal maturation is intimately linked with early endosomal fusion and therefore the flow of cargo through the endosomal system. Through the delineation of EEA1 endosomes into two distinct populations, namely N-terminally and C-terminally bound, we have shown that fusion of EEA1-bound vesicles is dependent on EEA1-cross binding between the two vesicles as has been evidenced previously[27,30]. Furthermore, we observe that this occurs only between vesicles that both contain EEA1 and that at least one endosome must be positive for PI(3)P, to enable stable C-terminal binding. EEA1 interacts with SNARE proteins Syntaxin 6 and Syntaxin 13 via the C-terminus of EEA1[53,54], which, post entropic collapse of EEA1, may execute the membrane fusion.

A relevant protein complex to this work is the mammalian class C core vacuole/endosome tethering (CORVET) system, which functions to mediate endosomal fusion independently of EEA1[55]. Surprisingly, overexpression of the N-terminal mutant of EEA1 also resulted in a similar phenotype of smaller, more fragmented APPL1 endosomes, with the exception that we found no APPL1-EEA1 double positive endosomes. It is unclear at what stage CORVET operates, and dissection of this question is beyond the scope of this study. However, the strong phenotype observed for the N-terminal mutant of EEA1 reinforces the role of EEA1 in self-regulating APPL1 to EEA1 conversion.

What is the physiological relevance of this mechanism? The trigger-and-convert approach provides emergent regulation of the

timing of early endosome maturation, leading to a tightly controlled and more timely and consistent flux of maturation, able to overcome the intrinsic stochasticity of single molecule protein–protein and protein–membrane interactions. This is critical to robust trafficking, as early endosomes act as stable sorting centres of endocytosed material, from which cargo is redirected towards the plasma membrane or sent to late endosomes and lysosomal degradation. As a result, robust maturation of cargo-bearing vesicles is a requirement of the intracellular transport system.

Furthermore, it has become increasingly apparent that many diverse transmembrane receptors are able to signal from within endosomes[56–61] and that signal attenuation may rely on trafficking to distinct intracellular destinations or organelles[62,63]. This suggests that the trafficking and maturation rate of endosomes is intrinsically coupled to the downstream signal transduction of transmembrane receptors[62,64,65], further highlighting the importance of tightly regulated intracellular transport itineraries that include transport and maturation[11,66]. An interesting corollary of this revised model of endosomal maturation is that we expect distinct very early endosomal populations to show different maturation times depending on their motility, which has consequences for rapidly versus slowly trafficked cargo as well as for statically anchored endosomes[11,67,68].

Our work highlights the power of rapid volumetric imaging, coupled with an unbiased analysis pipeline and complemented by simulations, to capture and describe dynamical processes and thus unravel mechanisms in unperturbed systems. Importantly, this approach precludes the need for genetic and pharmacological alterations that lead to the establishment of a new steady state or phenotype, thereby potentially obscuring the very dynamics that are to be studied. Emergent phenomena are central to biological processes across scales, and there are increasing evidence for structure–function relationships that extend far beyond molecular scales to form larger-scale patterns in space and/or time. In the endosomal system, the biochemical process of conversion is underpinned by phosphoinositide chemistry at the individual endosome level; at a population level, however, it is governed by the physical process of stochastic collisions that forms an inherent part of the transport system of endosomes. Importantly, this suggests that the robustness of the intracellular transport network may not derive solely from so-called 'master regulators' but through the complex dynamic interactions of individually noisy components to create emergent reproducibility of large-scale processes.

## Methods

### Cell lines
RPE1 and HeLa EEA1 knockout (KO) cells were incubated at 37 °C in 5% $CO_2$ in high glucose Dulbecco's modified Eagle's medium (DMEM) (Life Technologies), supplemented with 10% foetal bovine serum (FBS) and 1% penicillin and streptomycin (Life Technologies). Cells were seeded at a density of 200,000 per well in a six-well plate containing 25 or 5 mm glass coverslips.

### Live cell imaging
Cells were imaged using a lattice light-sheet microscope (3i, Denver, CO, USA). Excitation was achieved using 488 and 560-nm diode lasers (MPB Communications) at 1–5% AOTF transmittance through an excitation objective (Special Optics ×28.6 0.7 NA 3.74-mm immersion lens) and detected by a Nikon CFI Apo LWD ×251.1 NA water immersion lens with a ×2.5 tube lens. Live cells were imaged in 8 mL of 37 °C-heated DMEM and images were acquired with ×2 Hamamatsu Orca Flash 4.0 V2 sCMOS cameras.

### Plasmids and transfection
Wild-type HeLa cells were transfected with pEGFPC1-human APPL1, a gift from Pietro De Camilli (Addgene plasmid #22198)[6]; EEA1 TagRFP-T,

a gift from Silvia Corvera (Addgene plasmid #42635)[24]; EGFP-EEA1, a gift from Silvia Corvera (Addgene plasmid #42307)[13]; EGFP-Rab5, a gift from Marci Scidmore (Addgene plasmid #49888); and mRFP-Rab5, a gift from Ari Helenius (Addgene plasmid #14437)[69]. Cells were transfected with a total of 1 µg DNA (0.3 µg + 0.3 µg plasmid of interest + 0.4 µg blank DNA) using lipofectamine 3000 (Thermo Fisher Scientific). The DNA sequence for the N-terminal mutant of EEA1 carrying F41A and I42A, deficient in Rab5 binding, was synthesised and cloned into the TagRFP-T vector using the XhoI/BamHI sites. For the C-terminal binding mutant carrying R1375A, the synthesised sequence was cloned into the TagRFP-T vector using the XhoI/BamHI sites. It has been reported that drastic over-expression of APPL1 or EEA1 results in colocalisation of APPL1 and EEA1 on Rab5 endosomes; we, therefore, optimised this concentration by screening for this artefact and choosing conditions where we observed no overlap of APPL1 and EEA1.

### SiRNA INPP4A
RPE1 cells were transfected with APPL1-EGFP, EEA1-TagRFP, and either 10 nM INPP4A siRNA (AM16810, Thermo Fisher Scientific) or Silencer Negative Control siRNA (AM4611, Thermo Fisher Scientific) using lipofectamine 3000. ~24 h later the cells were imaged using epifluorescence microscopy (configuration as above). The cells were imaged sequentially with 100 ms exposure and at a rate of 3 s/frame for 20 min. The whole cell number of conversions within this window was reported for each condition.

### Fluorescence lifetime imaging
RPE1 cells were transfected with either EGFP-EEA1 + mRFP-Rab5, EEA1 TagRFP-T + EGFP-Rab5, EEA1-NTmut TagRFP-T + EGFP-Rab5 or EEA1-CTmut TagRFP-T + EGFP-Rab5 and either fixed with 4% paraformaldehyde or imaged live. The cells were imaged using an SP8 Falcon (Leica Microsystems) with an ×86 1.2 NA objective. Fluorescence lifetime images were acquired upon sequential excitation at 488 and 560 nm using a tuneable pulsed white-light laser at 10% transmission, with emission collected at 500–550 and 580–630 nm, respectively, using two Leica HyD detectors. The GFP lifetimes were fitted using a two-component fitting with $\tau_1 = 1.006$ ns and $\tau_2 = 2.600$ ns. The fixed images were analysed with pixel-wise lifetime fitting, and the live movies were analysed by separating the images into the two contributing fluorescence lifetime images.

### Drug addiction
Cells were incubated with 100 nM nocodazole in 8 mL DMEM for 5 min before and during imaging as indicated. Cells were similarly treated with 100 nM phorbol 12-myristate 13-acetate (PMA) (P1585, Sigma-Aldrich) 5 min before and during imaging as indicated. To selectively inhibit Vps34, cells were treated with 100 nM SAR405 (533063, Sigma-Aldrich) for 2 h prior to imaging and throughout the experiment.

### PI(3)P staining
To visualise PI(3)P localisation in relation to EEA1, immunofluorescence staining was performed as described previously[44]. Briefly, RPE1 cells were transfected with EGFP-EEA1 and mRFP-Rab5 and fixed in 2% PFA. These cells were then permeabilised using 20 µM digitonin for 5 min and labelled with 8 µg/mL recombinant GST-2xFYVE[70] which was detected using a GST primary antibody at 1:500 dilution(71–7500, Invitrogen) and a Goat anti-Rabbit AlexaFluor647 secondary antibody (A-21245, Thermo Fisher Scientific) at 1:500 dilution. These cells were then imaged using an SP8 Falcon as above, with the PI(3)P being detected using 647 nm excitation and emission collected at 660–700 nm, using a Leica HyD detector.

### Super resolution by radial fluctuations (SRRF)
RPE1 cells transfected with APPL1-EGFP and EEA-T-TagRFP were stimulated with 100 nM PMA as detailed above. The cells were then

imaged using widefield fluorescence microscopy with a Nikon Ti-2E body, ×100 1.5 NA objective (Olympus), and Prime 95B camera (Photometrics). Images were captured in 100 frame bursts with 5 ms exposure for each channel sequentially every 2 s for ~1 min image periods. The images were then processed using the SRRF plugin for Fiji[38,71].

### Segmentation and tracking analysis

Datasets analysed consisted of LLSM six movies of untreated and two movies of nocodazole-treated RPE1 cells. Images were first deskewed, then adaptive histogram equalisation and a median filter applied prior to blob detection using the Laplacian of Gaussian operator. The expected range of object sizes was supplied as an independent parameter for each fluorescence channel, with other parameters tuned to return a preliminary set of over-detected blobs, defined by centres of mass and approximate radii. From these data, representative regions denoting endosomes and background, respectively, were chosen from each movie in an unsupervised manner (by choosing the brightest and dimmest blobs, respectively); these regions were then used as templates to calculate cross-correlations against each candidate endosome. The results of this operation define a set of features for each object, which were used as inputs to a k-means clustering algorithm to classify objects into endosomes versus background (Supplementary Fig. 2).

A custom tracking routine built on trackpy[25] was then used to link objects into complete trajectories, independently for each channel. Trackpy is a package for tracking blob-like features in video images and analysing their trajectories, which consists of a Python implementation of the widely used Crocker–Grier algorithm[72] to link features (here, both localisation and intensity information) in time. Events of interest were then calculated by trajectory analysis, as follows. Correlated trajectories were classified as potential conversions[11,66], with stringent filters applied to exclude any events not clearly representative of APPL1 to EEA1 conversions (Supplementary Fig. 3a). To identify heterotypic collisions, local trajectories of neighbouring APPL1–EEA1 pairs were used to calculate the pairwise inter-endosome distance (separation between surfaces of nearby APPL1 and EEA1 endosomes along the line connecting their centres of mass). Local minima in the inter-endosome distance below a threshold value (within 200 nm, or roughly two pixels of overlap in the lateral dimension) were classified as collisions. These values were subsequently filtered to ensure that conversion-like events were excluded from the set of heterotypic collisions (Supplementary Fig. 3b). Events showing APPL1 to EEA1 conversions were classified as fusions or conversions, respectively based on whether or not the particular EEA1 track existed prior to colocalisation with APPL1 (Supplementary Fig. 3c). Events were classified as collision-induced versus unaided based on whether the APPL1 endosome collided with any EEA1 endosome in the 30 s prior to the event (Supplementary Fig. 3d).

### Photo-activated localisation microscopy (PALM)

Dendra-2 EEA1 was generated by replacing TagRFP-T in RagRFP-T EEA1 (Addgene plasmid #42635) at cloning sites AgeI and XhoI. Cells transfected with Dendra-2 EEA1 were fixed using 0.2% glutaraldehyde and 4% PFA in cytoskeletal buffer (10 mM MES, 150 mM NaCl, 5 mM EDTA, 5 mM glucose, and 5 mM MgCl$_2$) for 15 min at room temperature. The cells were washed gently three times with PBS. PALM microscopy was carried out with a Nikon N-STORM microscope with a ×100 oil immersion objective (1.49 NA) with a cylindrical lens for 3D localisation. A 488-nm laser beam was used for preconverted Dendra-2 excitation, with 405 nm for photoconversion and a 561-nm beam for post-photo converted Dendra-2. Localisations were exported to ViSP for visual examination and generating depth colour-coded images[73].

## Simulations

The endosome's surface was simulated as a bi-layered Fibonacci Sphere (a spherical grid in which neighbouring points are approximately equidistant). One layer consisted of Rab5 and the other PI(3,4)P2 or PI(3)P. The agents (APPL1, INPP4A, and N- and C-terminally attached EEA1) were allowed to stochastically attach and detach according to the schematic shown in Fig. 5a. The attachment rates increased with the number of neighbouring agents of the same type (cluster attach), and detachment rates increased with the number of neighbouring agents of the same type that detached recently (cluster detach). In addition, INPP4A had a fixed probability of converting PI(3,4)P2 to PI(3)P and a fixed probability of jumping to another nearby free PI(3,4)P2 after conversion. Upon collision, a fixed number of EEA1 N-terminal attachments were added to the endosome according to the availability of free Rab5 in a short time window after the collision. The conversion time of the endosome was measured as the first passage time of the fraction of PI(3,4)P2 converted to PI(3)P crossing a fixed threshold (set at 60%). See Supplementary Note 1 for details.

## Reporting summary

Further information on research design is available in the Nature Portfolio Reporting Summary linked to this article.

## Data availability

All LLSM, FLIM and imaging data will be available from the corresponding authors upon request. Supplementary movies are linked to this article. Source data are provided with this paper as a source data file. Source data are provided with this paper.

## Code availability

Codes may be accessed at the GitHub repository https://github.com/EndoMAT. (https://doi.org/10.5281/zenodo.8141082).

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

## Acknowledgements

The experimental work presented here was partially supported by the National Health and Medical Research Council of Australia (APP1182212) and ARC LIEF (LE150100163). H.M.Y. is supported by an Australian Government Research Training (RTP) Scholarship. A.P. and U.K.M. are supported by Monash Biomedicine Discovery Institute Scholarships. The EMBL Australia Partnership Laboratory (EMBL Australia) is supported by the National Collaborative Research Infrastructure Strategy of the Australian Government. The authors gratefully acknowledge the Imaging, FACS and Analysis Core and Cameron Nowell at Monash Institute of Pharmaceutical Science for their instrumentation and technical support. S.I.-B., K.J., and C.S.W. thank the Purdue Research Foundation, Purdue University start-up funds, and the Ross-Lynn Fellowship award for financial support. S.A. thanks Marino Zerial for the kind gift of HeLa EEA1 KO cell line. S.A. acknowledges John Carroll for critical inputs on the manuscript. S.A. and S.I.-B. thank all members of the Arumugam and Iyer-Biswas groups who contributed to discussions and gave feedback on the manuscript.

## Author contributions

S.A. conceived the study, designed the experiments and oversaw all aspects of the project; S.I.-B. directed the theory efforts, simulations, and the development of the automated analysis pipeline; H.M.Y., U.K.M. and S.A. performed LLSM experiments, H.M.Y. performed fluorescence lifetime experiments; S.J.R., C.A.M. and H.M.Y. performed PI(3)P staining experiments; H.M.Y. and S.A. performed super-resolution experiments; H.M.Y., K.J., C.S.W., H.G., A.P., L.Z.K. and S.A. analysed the data; K.J. performed agent-based simulations to propose, verify, and fine-tune mechanism under the guidance of S.I.-B.; C.S.W. developed the automated detection, tracking, and analysis pipeline and workflow under the guidance of S.I.-B.; S.A., H.M.Y., U.K.M., and H.G. contributed to molecular biology; and H.M.Y., K.J., C.S.W., S.I.-B. and S.A. discussed results, helped shape the research and analysis, and wrote the manuscript.

## Competing interests

The authors declare no competing interests.
