## [Peer Review File · Nature Communications]

Deterministic Early Endosomal Maturation Emerges From a Stochastic Trigger-and-Convert MechanismReviewers' Comments:

Reviewer #1:

Remarks to the Author:

In this article York and co-workers study early endosomal maturation and is proposed that a trigger-and-convert mechanism controls this process. Advanced imaging approaches have been applied both on live and fixed cells together with simulations. Very early endosomes APPL1 positive, interact with early endosomes (EEA1 positive) and therefore endosomal maturation is mediated by these heterotypic EEA1 interactions.

From the biophysics of fusion there is not enough detail explaining how the process of endosome-endosome fusion is driven and which molecules (fusogens?) are responsible of this. Is ESCRT III playing a role here? It is simple the interactions between EEA1 and APPL1 containing endosomes enough to drive the fusion reaction? More details about the fusion reaction between endosomes would be required to put into context this process.

In a relatively old paper the Zhuang group [Lakadamyali et al., 2006, Cell] postulate the existence of two types of early endosomes static and motile. Does this new model to maturation explain these results also? Are early endosomal compartments static the ones that did not undergo heterotypic interactions and failed to fuse? Please explain.

For the live cell (lattice light sheet) experiments [Figure 1]:

Are these cells stably expressing APPL1-EGFP and TagRFP-T EEA1?

How did you validate that function was not disrupted with the FP labelling?

Also, what about the endogenous population of unlabelled APPL1 and EEA1?

In regards the tracking algorithm used in the microscopy movies it would be worth to briefly define the main characteristics of the tracking software, even if it was published previously elsewhere...

Also, The intensity values are given in relative grey levels? Did you obtain the S factor of your camera to recover the data in number of photons? I assume that all acquisitions were taken with the same settings in terms of laser power, camera exposure, optical settings...so that all tracks can be compared from different acquisitions....

Can you please define first what is a cluster, how many molecules per cluster in each endosomal compartments were found and then also define the concept of counter-clustering?

Is there a particular stoichiometry in this process? Can you characterize this and relate it with the FRET FLIM data (see below)

In regards the FLIM experiments, please comment:

Fixing cells impacts FLIM lifetimes.

Also, the PNR endosomes might have a higher concentration as compared to endosomal compartments in the periphery of the cells. Plot Donor/Acceptor vs Lifetimes and show that over-expression is not a problem.

In regards the live cell FLIM experiments, perhaps the fitting approach is not the best. To use previous data on the donor and acceptor lifetimes coming from fixed cells might be a problem. First you need to demonstrate that fixing cells will not have an impact in the lifetimes of the FPs under study. Also,

there are other FLIM analysis methods that might be more useful with limited photons budget (such as the average lifetime and the phasor plot). Please apply these methods and compare your results. Remember to also plot the Donor/Acceptor intensity ratios versus the lifetimes for both the negative and positive results to demonstrate that the interactions that you are characterizing are real and not coming from over-expression.

"C-terminally bound EEA1, where the N-terminus is at least 150 nm away, extended into the cytoplasm from the Rab5 RFP" How this distance was calculated? Discuss

Transient transfection is known to disrupt endosomal traffic but this instance is not discussed in the manuscript. Stable cell lines should be produced and the impact of the endogenous protein assessed. Without checking for this, the whole paper is not validated.

Reviewer #2:

Remarks to the Author:

This is an interesting paper that carefully dissects the maturation of very early endosomes using state-of-the-art microscopy techniques. It provides advances on the role and mode of action of EEA1 on endosomal maturation. Some approaches need to be validated more and I have some suggestions for this. The paper can probably be written in a more concise and simple manner and the caveats more transparently disclosed. Rationales should be developed more.

Specific points:

Major

- 1) Rationales. At several locations, hypothesis or conclusions are made but the rationales behind these are not clearly explained. For example, in the abstract and in the text, it is stated that phosphoinositide modulation on endosomes drives endosomal maturation in a **stochastic** manner. I would rather interpret this as a regulated non-stochastic phenomenon. What is the rationale of saying this is a stochastic event? A reference is given in the introduction to emphasize stochasticity in the dynamics of vesicular transport system, but this reference corresponds to a modelling paper and not a research paper providing causal demonstration of stochasticity in endosomal maturation. See minor point 11 for another case of unexplained rationale.
- 2) These cells used in most experiments are EEA1 knock-out cells transfected with the indicated constructs.
 - a. Why not using wt cells as knocking out EEA1 will perturb the endocytic machinery and it is unclear how quickly this is restored by expression of ectopic EEA1.
 - b. I note that the EEA1 vesicles are big, probably bigger than in control cells. Is this the case?
 - c. This work would be strengthened if key experiments are reproduced with cells that express endogenous levels of tagged EEA1 and APPL1. This requires knock-in the APPL1-EGFP- and TagRFP-T-tagged constructs in the endogenous loci.
- 3) The authors have designed a pipeline to detect collisions, conversions, and fusions. This pipeline should be thoroughly tested and validated as it is at the core of the experimental paradigm used by the authors in this study. What happens when the routine is used on two vesicular structures that are not expected to interact such as LAMP-1-positive vesicles and APPL1-positive endosomes? I have concerns with the detection of collisions in particular. Two vesicles getting closer to one another do not necessarily "collide" so that they can exchange material. Is there a way to demonstrate transfer of material between two vesicles?
- 4) Collision detection: based on what I can see with the automation procedure, endosomes are given a defined radius. This radius appears quite large to me. Enlargement of endosomes might affect detection of actual collisions. If the radius of endosomes is reduced during the automation procedure, are similar results obtained?
- 5) Kinetics of marker intensities (as for example in Figure 1c-d). Intensities of endosomal markers are

measured as a function of time in several panels. Is this intensity determined on the same light-sheet plane throughout the experiment or is the light-sheet plane adjusted in in each frame so as to go through the centroid of *each* endosome analyzed on the image? If the first option was chosen, intensity variations may result from endosomes moving in and out of the plane.

6) Line 147: "true endosomes". These "true endosomes" are defined based on fluorescence intensities. Can this be validated somehow? Could the authors use a series of organelle markers (e.g., ER, mitochondria, Golgi markers) and check that their routine does not recognize these as "true endosomes"?

7) In Supplementary Figure 3, the terminology used in the workflow is not easy to grasp. It is difficult to understand what terms like "loose tracking criteria" mean as they are not precisely defined. This brings up a general comment: the authors should try to explain their procedures in simpler terms, and they should disclose the actual values of the parameters used in their procedures. Moreover, these parameters should be clearly defined.

8) Figure 4b. Additional controls would be warranted. What is the impact of SAR405 and siRNA INPP4A on PI(3)P levels? How efficient is the siRNA against INPP4A?

9) Figure 6. This model is confusing and does not reflect very well the findings of the authors. In my opinion, the most salient observation is that EEA1 is acquired first by early endosomes through its N-terminal moiety. Then, as endosomes mature, additional EEA1 molecules are recruited via their C-terminal moiety. This is what the model should concentrate on. The other aspects are much more speculative and should be disclosed as such. For example the authors take for granted that collisions allow endosomes to exchange materials but this is not shown in the manuscript. Why is the possibility of unbound, cytoplasmic EEA1 acquisition not considered here? Since APPL1-positive endosomes were observed acquiring EEA1 without collisions, shouldn't this possibility be considered? Why is EEA1 observed binding a single endosome via both its N- and C-terminal domains? It is mentioned earlier on that such interaction is not possible due to the EEA1's rigid quaternary structure. Why is there a full switch between N-terminal and C-terminal EEA1 binding in the model? Why not a gradual acquisition of C-terminal binding on the areas of the endosomes that become PI(3)P positive? How can endosome 1 "steal" EEA1 from endosome 0?

In line 389, the authors state that "collisions between endosomes form an important step in overall endosomal conversions rates". I do not think the data permit reaching this conclusion. As indicated above, characterizing collisions is prone to many caveats. There may be a correlation between endosomes having experienced collisions but whether these putative collisions increase the conversion rate is one possibility among others. One could argue that it is the velocity of the endosomes through their interactions with microtubules and motor proteins that facilitate their conversion, not the bouncing of endosomes on other vesicles (faster moving endosomes are more likely to encounter other vesicles but these encounters may play no role in endosome maturation). In conclusion, I feel that the authors should tune down some of their claims and, in their model, concentrate on their most solid findings.

Minor

1) I would refrain from using normative words that are poorly meaningful in biology such as "robust".

2) What is the usefulness of mentioning dynein in the introduction and raising questions related to this protein as the paper is not directly addressing the role of dynein in endosomal maturation?

3) Line 128: "To simultaneously measure *ensemble* endosomal conversion dynamics". I do not understand this sentence.

4) Supplementary movie 4. This movie should be "labelled" so that we can follow the authors' interpretation. For example, in the control cells, where are all the conversion events? It would be nice to have them indicated by labels.

5) Line 208: "were also "pulsatile" suggesting evidence of clustering". What is the rationale of this statement? Couldn't this "pulsatility" due to vesicles moving in and out of the light sheet?

6) Line 219: supplementary figure 6. To which structures are all the little dots corresponding to?

7) Line 224 and 226: the figures are not supplementary.

8) Line 257: Fig. 2c. We do not see the nucleus in the figure. Please label it.

9) Line 349: Supplementary Figure 9. This figure lacks a control (untreated cells)

- 10) Is there a repository available for all the collected data used in the article?
- 11) Supplementary Figure 5c. I do not understand this figure. What does it show? How can it "hint at periodicity"?
- 12) Figure 1g. The clustering of APPL1 and EEA1 should be quantitated somehow. At present, only one frame is shown and only one vesicle was analyzed over time.
- 13) Figure 2c. Was the RFP signal obtained solely through FRET or was RFP signal obtained after illuminating the RFP fluorophore with the RFP excitation wavelength?
- 14) In Supplementary movie 6, shouldn't we see the blue signal (N-terminally-bound EEA1) eventually being replaced by the green signal (C-terminal-bound EEA1)? This does not appear to be the case in this movie.
- 15) Figures 4b and 4d. Please describe precisely how the boxplots were constructed. What do they show? Same comment for the violin plots. What are the diamonds in the violin plots?
- 16) Line 380: "the localisation of this [R1375A] mutant was largely cytosolic". Isn't this unexpected as the first binding occurs via the other end (the N-terminus)? This mutant should therefore be seen on very early endosomes.
- 17) There is an inconsistency between what is stated in line 430 ("the N-terminus of EEA1 has weak binding affinity to Rab5") and in line 482 ("The N-terminus forms the stronger Rab5 interaction").
- 18) Figure 1b. Doesn't **a** fuse with **b**? If not where is **a** after the "collision". In this figure, it would help the readers that endosomes **a** to **d** are indicated on all the images where they are found (from their initial detection to their eventual disappearance). This comment is valid for other similar figures where endosomes are labelled.
- 19) Figure 1d (legend). Typo: "endosome" not "endsome".
- 20) Figure 4b (graph). For clarity I suggest to write "APPL1 to EEA1 conversion" and "Fusion between APPL1 and EEA1 vesicles" directly on the figure.

Reviewer #3:

Remarks to the Author:

This manuscript used a battery of cutting-edge, fluorescence based quantitative imaging approaches to build a model of very early endosome maturation.

Lattice light sheet microscopy allows complete volumetric imaging of RPE cells expressing fluorescent APPL1 and EEA1 endosomal proteins at diffraction-limited and ~second time intervals over extended periods (many minutes). Combined with an unbiased, automated segmentation and tracking pipeline, the manuscript is able to produce a pan-optical reconstruction of the maturation process, which produces the first surprise in how maturation occurs: although the authors do observe occasional spontaneous conversion of APPL1 endosomes into EEA1 endosomes (12%), these are usually preceded by one or more collisions with EEA1 endosomes (11%) or else occur by fusion with existing EEA1 endosomes (77%). Using FLIM of an N-terminally tagged EEA1 with a tagged Rab5, the manuscript can differentiate EEA1 bound to Rab5-positive endosomes via their N- or C-termini. Curiously, fusion is only observed to occur between EEA1 labelled endosomes where at least one endosome is attached by C-terminally anchored EEA1, whereas maturation involves a transition from N-terminally to C-terminally dominated EEA1. The endosomal lipid PI(3)P is predominantly associated with c-terminally attached EEA1 as expected. Inhibition of the major PI(3)P-synthesizing VPS34 enzymes abolished endosome fusion, but not conversion - whereas blocking PI(3,4)P2 to PI(3)P conversion using INPP4A siRNA blocks both processes - implying a PI(3,4)P2 to PI(3)P switch is essential to both modes of maturation. An N-terminal mutant of EEA1 unable to bind Rab5 mildly inhibits fusion but not conversion in RPE1 cells, but cannot restore maturation at all in EEA1 KO HeLa cells. The observations are synthesized into a unifying model of endosome maturation: an initially PI(3,4)P2, Rab5 and APPL1 coated very early endosome collides with c-terminally anchored EEA1 endosome, which allows n-terminal binding of Rab5. This interaction displaces APPL1, allowing further EEA1 binding. This cascades until such a time that enough APPL1 has been displaced to free Rab5 binding sites and for INPP4 to convert the lipid to PI(3)P, which allows either maturation by c-terminal binding of EEA1 or

else fusion with more mature endosomes.

Overall, the manuscript is technically impressive and produces a novel model for endosome maturation that is still by and large consistent with prior literature. The manuscript is clearly written, and the data are beautifully presented and convincing. I believe the manuscript will be an important vertical advance in the field of membrane traffic, with the approach and concepts applicable to other trafficking processes. The manuscript is therefore of significant interest to a wide swathe of the biological and biomedical sciences, and appropriate for publication in Nature Communications (or Nature Cell Biology, honestly - that would be a better home in my opinion).

Overall, the reviewer found no major flaws. However, there is one area that could be improved. Specifically, the sections describing "Endosomal conversions are driven by phosphoinositide conversions by INPP4A" seemed overly simplistic to me:

Firstly, although SAR405 barely inhibits conversion of VEEs compared to INPP4A siRNA, it is rather more effective at blocking fusion. However, this is not commented on in the text, despite the fact that it would make sense as PI3P is more prevalent on the more mature, C-terminally EEA1-bound early endosomes.

Secondly, the manuscript lacks clear controls for the extent of PI(3)P depletion by these two treatments. Data should be included documenting the extent of PI(3)P depletion by 100 nM SAR405 or INPP4A knock-down, using the GST-2xFYVE probe utilized in other experiments.

Thirdly, PI(3,4)P₂ conversion to PI(3)P being the source of this lipid required for the conversion process is intriguing, and could imply that the lipid ultimately comes via the PI3KC2A pathway; this can now be directly tested with highly potent and selective PITCOIN inhibitors of these enzymes (doi: 10.1038/s41589-022-01118-z). It would be interesting and informative (but not essential) to this model to test whether these inhibitors also block conversions and fusions.

Fourthly, the fact that INPP4A inhibition and VPS34 inhibitors blocks fusion is curious and unexpected; it implies that lipid is required on both endosomes (VPS34-derived on later early endosomes, and INPP4A-derived on VEEs). However, this is not addressed in the manuscript.

Lastly, the model discussed at the end of the manuscript describes a direct interaction of APPL1 with PI(3,4)P₂. However, my recollection of the literature is that this is essentially implicit from the co-incident timing of effectors and PI enzymes, coupled with poorly defined lipid binding by APPL1. However, I cannot recall explicit demonstrations of a specific interaction between APPL1 and PI(3,4)P₂. If there is one, this should be stated; if not, the implicit nature of the interaction should be spelled out in the manuscript.

I also spotted a couple of places where technical clarifications are required:

Figures 2b, 3a, b and d: it is not always clear what the data refer to, in terms of individual data points, box and whisker or violin plots. How do these relate to the number of endosomes, number of cells, and number of experiments? This information should be included in the legends. Likewise for the summary statistics included in the box plots - its not clear if these are means or medians etc.

REVIEWER COMMENTS

Reviewer #1 (Remarks to the Author):

In this article York and co-workers study early endosomal maturation and it is proposed that a trigger-and-convert mechanism controls this process. Advanced imaging approaches have been applied both on live and fixed cells together with simulations. Very early endosomes APPL1 positive, interact with early endosomes (EEA1 positive) and therefore endosomal maturation is mediated by these heterotypic EEA1 interactions.

From the biophysics of fusion there is not enough detail explaining how the process of endosome-endosome fusion is driven and which molecules (fusogens?) are responsible for this. Is ESCRT III playing a role here? It is simple the interactions between EEA1 and APPL1 containing endosomes enough to drive the fusion reaction? More details about the fusion reaction between endosomes would be required to put into context this process.

We have now added additional discussion of the consequences of this model on early endosomal fusion to the manuscript on lines 574–583. In summary, endosomal fusion is a complex and regulated process (Gautreau et al. 2014 *Cold Spring Harbor Persp*) which involves the specific action of a range of proteins including SNAREs (Dingjan et al. 2018 *Phys Revs*), as well as endosomal tethers such as EEA1, which is the longest tether molecule playing a central role in early endosomal size regulation (Christoforidis et al. 1999 *Nature*, Mills et al. 2001 *J Cell Sci*, Dumas et al. 2001 *Mol Cell*). EEA1 is a large ‘antenna’ like molecule that projects from the endosomal surface and is able to bind endosomes at both ends. Following binding at both the C- and N-terminus EEA1 undergoes an entropic collapse to help bring together the two membranes within a distance for other components of the fusion machinery to function (Murray et al. 2016 *Nature*). EEA1 interacts with SNARE proteins Syntaxin 6 and Syntaxin 13 via the C-terminus of EEA1 (Simonsen et al. 1999 *J Biol Chem*, McBride et al. 1999 *Cell*) which help execute membrane fusion. In this paper we have extended this model through the delineation of EEA1 endosomes into two distinct populations, namely N-terminally and C-terminally bound, we have shown that fusion of EEA1-bound vesicles is dependent on EEA1-cross binding between the two vesicles as has been evidenced previously (Dumas et al. 2001 *Mol Cell*, Murray et al. 2016 *Nature*). Furthermore, we observe that this occurs only between vesicles that both contain EEA1, and that at least one endosome must be positive for PI(3)P, to enable stable C-terminal binding.

To our knowledge, ESCRTIII has not been shown to play a role in early endosomal fusion (Vietri et al. 2020 *Nat Rev Mol Cell Biol*), but CORVET have been postulated to have a complementary, PI(3)P independent-role in fusion (Perini et al. 2014 *Traffic*). Owing to the complex organisation and dynamics, deciphering which biochemical compartments it affects in relation to PI(3)P-independent activity is yet to be studied in a dynamic manner. We have detailed this in the discussion in the manuscript on lines 609–613.

In a relatively old paper the Zhuang group [Lakadamyali et al., 2006, *Cell*] postulate the existence of two types of early endosomes static and motile. Does this new model to maturation explain these results also? Are early endosomal compartments static the ones that did not undergo heterotypic interactions and failed to fuse? Please explain.

Lakadamyali et al. demonstrated that early endosomes show distinct motility characteristics dependent on their internal cargo; for example, Lakadamyali et al. showed that EGF- and transferrin (Tf)-bearing endosomes show drastic differences in their motilities with EGF endosomes undergoing rapid trafficking to the perinuclear region where as Tf-bearing endosomes are more static. We have previously reported on the mechanism of this rapid EGF endosome trafficking that is triggered by EGF induced calcium currents, leading to APPL1 re-localization (York et al. 2021 *Commun Biol*).

Whilst the link between transport and maturation is a complex one, transferrin-containing endosomes undergo more constitutive trafficking acquiring APPL1 and maturing to EEA1 endosomes over a longer time period. In comparison, EGF acquires APPL1 almost immediately following receptor EGFR dimerisation and shows rapid directed motion towards the perinuclear region where it then matures into an EGF-EEA1 compartment. Consistent with the model proposed in this manuscript, we observed that inhibiting the motility of APPL1-EGF endosomes led to delayed endosomal maturation into EEA1-bound vesicles, leading us to suggest the original hypothesis that heterotypic interactions may be important. In the discussion we have made specific reference to the consequences of this model on motile and static endosomes on lines 599–607.

For the live cell (lattice light sheet) experiments [Figure 1]:

- Are these cells stably expressing APPL1-EGFP and TagRFP-T EEA1?
- How did you validate that function was not disrupted with the FP labelling?
- Also, what about the endogenous population of unlabelled APPL1 and EEA1?

These cells are transiently expressing a small amount of fluorescently labelled APPL1 and EEA1 in addition to the endogenous 'dark' proteins. To ensure we were only adding a low amount of additional protein to these cells we used only 30% of total DNA for expressing APPL1 and EEA1 and the rest being blank DNA. When screening for cells we employed two quantitative approaches to ensure we were imaging cells that did not display altered trafficking or conversion dynamics.

1. One of the effects of APPL1 overexpression is enhanced colocalization with Rab5 and EEA1, in agreement with Zoncu et al. 2009 *Cell*. In general, APPL1 and EEA1 only show very little colocalization in the absence of any artefacts. By co-expressing APPL1 and EEA1 supplemented with blank DNA, we found that only a small percentage of vesicles showed colocalization for APPL1 and EEA1 in a dynamic fashion pertaining to conversions as compared to overexpression, where we see stabilised double positive endosomes.
2. We rely on the direct mean intensity to ensure that we are picking cells that have a very small amount of fluorescently expressing proteins, resulting in minimal perturbation of the processes.

We have discussed these measures to avoid overexpression artefacts in the methods section.
Line: 652-656

The constructs we are using have been previously demonstrated to be functional and have been widely used in the literature (Zoncu et al. 2009 *Cell* Navaroli et al. 2011, *PNAS*, Lawe et al. 2000 *J Biol. Chem*).

In regards the tracking algorithm used in the microscopy movies it would be worth to briefly define the main characteristics of the tracking software, even if it was published previously elsewhere...

Trackpy is a package for tracking blob-like features in video images and analysing their trajectories; it consists of a Python implementation of the widely-used Crocker–Grier algorithm (Crocker and Grier 1996, *J Colloid Interface Sci*) and links features (here, centres of mass and fluorescence intensities) in time. Care was taken to choose parameters that result in continuous trajectories without spurious links being made (as verified by visual inspection of the trajectories overlaid onto the movies). These details have been updated in the Methods section on lines 725–731.

Also, The intensity values are given in relative grey levels? Did you obtain the S factor of your camera to recover the data in number of photons? I assume that all acquisitions were taken with the same settings in terms of laser power, camera exposure, optical settings...so that all tracks can be compared from different acquisitions....

Yes, all fluorescence intensity values are given in relative grey levels, calculated by subtracting the local background from the total signal intensity over each endosome. All acquisitions were taken with the same settings in terms of laser power, camera exposure, and optical settings, thus aiding in comparisons across different acquisitions. Note the following additional points. Firstly, with respect to image analysis routines used to identify and track endosomes, intensity values were normalised on a per-frame basis to standardise detection of endosomes (i.e., to ensure that numbers of detected endosomes do not change as a function of total fluorescence signal). Secondly, with respect to reporting of aggregated time series, fluorescence levels were normalised with respect to the events of interest (i.e., start of conversion) to facilitate cross-endosome comparisons.

Can you please define first what is a cluster, how many molecules per cluster in each endosomal compartments were found and then also define the concept of counter-clustering?

In these data acquired using PALM/SRRF, the higher resolution available allows for resolution of clusters based not solely on intensity values, but also on area (in the simplest case, the number of contiguous pixels of a specified fluorophore above a threshold intensity value). Various techniques for sophisticated clustering methods of super-resolution data have been developed (see Khater et al. 2020 *Patterns* for a review); a straightforward approach is to measure overlap in signal across multiple channels (for a detailed protocol, see the examples provided Roberts et al. 2018 *Bio Protoc*). In our super-resolution data, intensity values of

opposite channels are not observed to coincide. We term this lack of signal colocalization in the super-resolution data “counter-clustering”, indicating that we observe clusters (contiguous regions of fluorescence intensity) of either APPL1 or EEA1, but not both APPL1 and EEA1. We have now quantified this counter-clustering and have added a new supplementary figure (Supplementary Fig. 8).

Is there a particular stoichiometry in this process? Can you characterize this and relate it with the FRET FLIM data (see below)

We are unsure of precisely which molecules the reviewer is referring to here. With regards to APPL1 and EEA1 stoichiometry we observe that pre-conversion, APPL1 endosomes contain no discernible EEA1 signal as measured under a mask of APPL1 signal. Similarly, following conversion we observe total loss of APPL1 signal under the EEA1 mask channel (see Fig. 1g,h).

With respect to NT-bound EEA1 vs CT-bound EEA1 we can estimate the fraction of N- and C-terminally bound EEA1 by using the phasor plot to measure the quenched and unquenched donor lifetimes (which fits with the lifetimes measured from the EEA1 EGFP only cells) and measure a pixel-wise FRET efficiency (Response Fig. 1a). Using these values we estimate that peripheral endosomes are nearly entirely bound by molecules showing FRET (N-terminally bound EEA1), which is comparable to the FRET percentage measured in SAR405-treated cells which are expected to be largely N-terminally bound due to the loss of PI(3)P (Response Fig. 1b,c and Fig. 3a, Supplementary Fig. 14). This is in agreement with our observations in live-FLIM experiments where we observe initial EEA1 binding is always via the N-terminus (Fig. 3). In comparison, perinuclear endosomes show a range of ‘FRET percentages’ with a mean of ~25% photons being estimated to be detected as FRET photons, and with a sizable proportion of pixels showing very low (<5%) FRET.

Response Figure 1. Estimated pixel-wise FRET percentages in EEA1 EGFP Rab5 RFP expressing cells. (a) Representative FLIM image colour-coded for estimated FRET-percentages at each pixel ranging from 0% (*blue*) to 100% (*red*). Scale bar = 10 μ m. **(b)** Cumulative frequency histogram of perinuclear (*green*) and peripheral (*blue*) endosome FRET percentages. **(c)** Histogram of FRET percentages of perinuclear (*green*) and peripheral (*blue*) endosomes.

In regards the FLIM experiments, please comment:

- Fixing cells impacts FLIM lifetimes.
- Also, the PNR endosomes might have a higher concentration as compared to endosomal compartments in the periphery of the cells. Plot Donor/Acceptor vs Lifetimes and show that over-expression is not a problem.

See detailed response below.

In regards the live cell FLIM experiments, perhaps the fitting approach is not the best. To use previous data on the donor and acceptor lifetimes coming from fixed cells might be a problem. First you need to demonstrate that fixing cells will not have an impact in the lifetimes of the FPs under study. Also, there are other FLIM analysis methods that might be more useful with limited photons budget (such as the average lifetime and the phasor plot). Please apply these methods and compare your results. Remember to also plot the Donor/Acceptor intensity ratios versus the lifetimes for both the negative and positive results to demonstrate that the interactions that you are characterizing are real and not coming from over-expressiion.

To verify that the fixation protocol is not affecting our measured fluorescence lifetimes we measured the distributions of mean lifetimes of peripheral and perinuclear endosomes in live cells transfected with EEA1 EGFP and Rab5 RFP. In short, we observed almost identical mean lifetimes for each population; 1.91 and 2.12 ns for peripheral and perinuclear, respectively, compared with our measurement over many cells obtained from fixed cells (1.87 and 2.10 ns, respectively); see Response Fig. 2. This data has now been added to the supplementary information in Supplementary Fig. 11. This is agreement with a previous paper that carried out a systematic investigation of the effects of fixation on a range of fluorophore lifetimes (Joosen et al. 2014 *J Microsc*). Joosen et al. report that fixation by formaldehyde or methanol itself does not affect the lifetime of genetically encoded fluorescent proteins produced in cells.

Response Figure 2. Live-FLIM imaging of EEA1 Rab5 endosomes reproduces lifetime distributions consistent with fixed data. Normalised frequency histograms of the detected fluorescence lifetimes of EEA1-EGFP photons measured in peripheral endosomes (*blue*) and perinuclear endosomes (*green*) in live cells.

To corroborate our findings from live FLIM we have now used both average lifetime and phasor plot analysis as suggested by the reviewer. Using the phasor plot to select the region of FRET-ing pixels, we colour-coded the same N- to C-terminal conversion montage included in the paper (Fig. 3d,e). Consistent with our previous findings we observe a clear loss of FRET pixels concomitant with endosomal fusion and maturation. This is now included as Supplementary Fig. 11a,b. We also visualised and measured the average lifetimes of each pixel during this maturation and observed an increase in average lifetime as expected; this montage has now been included in Supplementary Fig. 11c,d. Given that phasor plots are most often used to generate binary maps of FRET or non-FRET pixels and that pixelwise average lifetimes can be quite noisy at lower photon counts, we have chosen to retain the two-component fit analysis that nicely captures the change in EEA1 orientation in the main text and have included these alternative analysis modes in the supplement for completeness.

We thank the reviewer for their helpful suggestions regarding Donor/Acceptor intensity ratio versus lifetime plots and overexpression. Firstly, to demonstrate that EEA1 orientation can lead to FRET or non-FRET with Rab5, we have now included FLIM data for EEA1 mutants that are defective in endosomal binding at either the N- or C-terminus (Supplementary Fig.

10). In these controls we can clearly observe only one lifetime for endosomes across the entire cell as a result of only one permissible binding; via the N-terminus in the CT-mutant corresponding to shorter lifetimes and via the C-terminus in the NT-mutant corresponding to longer lifetimes. Secondly, to visualise the relationship between donor/acceptor intensities and the corresponding lifetimes, we have included an example representative FLIM image where donor/acceptor ratio and fluorescence lifetime are colour coded; here, we can observe in one field of view that there are endosomes with each combination of donor/acceptor ratio and fluorescence lifetime, highlighting that this shortening of lifetime arises from changes in the FRET distance between donor and acceptor molecules due to differences in terminal binding, and not as a result of different donor/acceptor ratios. This has now been included as a supplementary figure (Supplementary Fig. 10).

Furthermore, we have now included the suggested donor/acceptor intensity versus lifetime plots for peripheral and perinuclear endosomes in cells expressing full-length EEA1, as well as plots for the CT- and NT- EEA1 mutants (Response Fig. 3b–e). In summary, the NT-EEA1 mutant shows only one (longer) lifetime. The CT-EEA1 mutant plot shows that a majority of pixels have a shorter lifetime but at high Donor/Acceptor Ratios there is a shift to a longer lifetime corresponding to pixels in which there is Donor (Rab5 EGFP) but little to no acceptor to quench the lifetime, hence producing a shift towards the non-FRETing lifetime. In the peripheral endosomes of cells expressing full-length EEA1, we observe a similar double population of majority FRETing pixels with a smaller proportion of non-FRETing pixels at high D/A ratios. In the case of perinuclear endosomes, we observe that the majority of pixels show a longer lifetime with a small proportion of FRETing pixels (unsurprisingly, as these endosomes may have some residual N-terminally bound EEA1 molecules); importantly however, the majority of pixels showed a longer lifetime corresponding to non-FRETing pixels even at low Donor/Acceptor ratios. To quantify this, we have performed a Gaussian fit of the lifetime values at both high and low donor acceptor ratios to show that the average fluorescence lifetime of endosomes is longer (less FRET is occurring) in perinuclear endosomes even at comparable donor acceptor ratios, mimicking what is observed in the NT and CT EEA1 mutants (Response Fig. 3f).

Response Figure 3. Fluorescence lifetime shift between peripheral and perinuclear endosomes is independent of donor/acceptor ratio. (a) Representative FLIM image of RPE1 cell expressing EEA1 EGFP and Rab5 RFP showing donor acceptor intensity ratio (left) and fluorescence lifetime (right). Example endosomes are numbered: #1 has both low D/A ratio and lifetime, #2 has both high D/A ratio and lifetime, #3 has low D/A ratio but high lifetime, and #4 has high D/A ratio but low lifetime. Scale bar = 2 μ m. (b–e) Density-coded scatterplots of Donor/Acceptor intensity ratio (Log10) plotted against donor fluorescence lifetime, for peripheral endosomes (b) and perinuclear endosomes (c) in cells expressing EEA1 EGFP

and Rab5 RFP, or for all endosomes in cells expressing Rab5 EGFP and EEA1 CT-Mut TagRFP **(d)** or EEA1 NT-Mut TagRFP **(e)**. Donor/Acceptor Ratios above 1 have been coloured red and ratios below 1, blue. **(f)** Table of mean donor lifetimes \pm standard deviation of Gaussian distributions fit to the top or bottom half of donor/acceptor ratios.

“C-terminally bound EEA1, where the N-terminus is at least 150 nm away, extended into the cytoplasm from the Rab5 RFP” How this distance was calculated? Discuss

This value was not calculated but is instead hypothesised from known literature. Murray et al. 2016 *Nature* showed using electron microscopy that EEA1 extends as a rigid molecule from the endosomal membrane \sim 200nm (measured 141 ± 47 nm). This is in agreement with molecular dynamics simulations which investigated the structure of EEA1 bound to endosomal membranes (Larson et al. 2021 *PLoS Comp Biol*). Given that this is far greater than the FRET distance <10 nm we reasoned that the only FRET interaction permissible was via NT binding to Rab5, as the GFP was tagged to the N-terminal of EEA1. This is supported by the evidence of two distinct fluorescence lifetime populations when full-length EEA1 is expressed, or the single lifetime populations observed when EEA1 mutants with reduced binding are expressed (detailed above).

Transient transfection is known to disrupt endosomal traffic but this instance is not discussed in the manuscript. Stable cell lines should be produced and the impact of the endogenous protein assessed. Without checking for this, the whole paper is not validated.

To our knowledge there has been no report of lasting endosomal trafficking disruption due to the various methods of transient transfection as compared to stable overexpression methods. However, overexpression of endosomal proteins that are often found at low concentrations in endogenous conditions can indeed lead to altered trafficking dynamics.

To ensure we were only adding a low amount of additional protein to these cells we used only 30% of total DNA for expressing APPL1 and EEA1 and the rest being blank DNA. When screening for cells we employed two quantitative approaches to ensure we were imaging cells that did not display altered trafficking or conversion dynamics.

1. One of the effects of APPL1 overexpression is enhanced colocalization with Rab5 and EEA1, in agreement with Zoncu et al. 2009 *Cell*. In general, APPL1 and EEA1 only show very little colocalization in the absence of any artefacts. By co-expressing APPL1 and EEA1 supplemented with blank DNA, we found that only a small percentage of vesicles showed colocalization for APPL1 and EEA1 in a dynamic fashion pertaining to conversions as compared to overexpression, where we see stabilised double positive endosomes.
2. We rely on the direct mean intensity to ensure that we are picking cells that have a very small amount of fluorescently expressing proteins, resulting in minimal perturbation of the processes.

We have made specific note of this in the methods section on lines 652–656. “It has been reported that drastic over-expression of APPL1 or EEA1 results in colocalization of APPL1

and EEA1 on Rab5 endosomes; we therefore optimised this concentration by screening for this artefact and choosing conditions where we observed no overlap of APPL1 and EEA1.”

Reviewer #2 (Remarks to the Author):

This is an interesting paper that carefully dissect the maturation of very early endosomes using state-of-the-art microscopy techniques. It provides advances on the role and mode of action of EEA1 on endosomal maturation. Some approaches need to be validated more and I have some suggestions for this. The paper can probably be written in a more concise and simple manner and the caveats more transparently disclosed. Rationales should be developed more.

Specific points:

Major

1) Rationales. At several locations, hypothesis or conclusions are made but the rationales behind these are not clearly explained. For example, in the abstract and in the text, it is stated that phosphoinositide modulation on endosomes drives endosomal maturation in a stochastic manner. I would rather interpret this as a regulated non-stochastic phenomenon. What is the rationale of saying this is a stochastic event? A reference is given in the introduction to emphasize stochasticity in the dynamics of vesicular transport system, but this reference corresponds to a modelling paper and not a research paper providing causal demonstration of stochasticity in endosomal maturation. See minor point 11 for another case of unexplained rationale.

Phosphoinositide modulation on endosomes as well as appearance of proteins involved in maturation as measured by their intensities support the idea of stochasticity in this system (Hansen et al. 2019 *PNAS*). Further, the molecular number in the process of endosomal maturation and the time dependent variations observed are interpreted to be consistent with a stochastic model (Puchner et al. 2013 *PNAS*).

Endosomal PI modulation as measured experimentally and modelled are discrete and few in number over time, as well as individual events also comprise of low number of participating molecules. Conversion mechanisms in Rab5 and Rab7 are shown to be bistable and stochastic (Rink et al. 2006 *Cell*, Bezeljak et al. 2020 *PNAS*). These experimental papers and additional modelling papers interpret that the processes that build endosomal network—trafficking, sorting and maturation are stochastic events (Vagne and Sens 2018 *Biophys J*, Vagne and Sens 2018 *Phys Rev Lett*, Bressloff and Newby 2013 *Rev Mod Phys*). This is the basis of our starting point, where we address if there are mechanisms to overcome stochasticity and the endosomal maturations ‘appears’ to be deterministic.

Individual events are stochastic in nature, and without some specific interactions in place, the system will continue to be stochastic and will not go forward. This is what we see when we perturb the cells with Nocodazole—some APPL1 endosomes transiently acquire EEA1, but never fully convert (supplementary Movie 4). We are describing a mechanism which increases the likelihood of these stochastic interactions which then gives rise to an emergent

deterministic process. These interactions being the act of inter-endosomal collisions that result in a regulated, deterministic outcome out of initially stochastic conditions.

2) These cells used in most experiments are EEA1 knock-out cells transfected with the indicated constructs.

a. Why not using wt cells as knocking out EEA1 will perturb the endocytic machinery and it is unclear how quickly this is restored by expression of ectopic EEA1.

This is a misunderstanding; the majority of the experiments were carried out in wild-type RPE1 cells which are mildly overexpressing fluorescent APPL1 and EEA1. The EEA1 knock-out cells were only used in the rescue experiments (Fig. 4c,d).

b. I note that the EEA1 vesicles are big, probably bigger than in control cells. Is this the case?

This is not the case; EEA1 endosomes have been shown to have a range of sizes that can be further influenced by cell type, cell state and mechanism of endocytosis. In our RPE1 LLSM data we measured the mean diameters of EEA1 vesicles as $0.84 \pm 0.16 \mu\text{m}$ (mean and SD); a similar mean was measured in the HeLa EEA1 KO cells transfected with full length EEA1. This matches fairly well to the size of endogenous EEA1 vesicles as measured by CLEM in A549 (Van der Beek et al. 2022 *J Cell Biol*) of $\sim 0.4 \mu\text{m}$ diameter, when accounting for the diffraction-limited point spread function of $0.25\text{--}0.3 \mu\text{m}$ which convolutes the ground truth. Other diffraction-limited imaging of EEA1 vesicles show a similar size of $0.8\text{--}1.0 \mu\text{m}$ diameter (Wilson et al. 2017 *MBoC*, Vonderheit and Helenius 2005 *PLoS*, Paramasivam et al. 2022 *J Cell Biol*).

c. This work would be strengthened if key experiments are reproduced with cells that express endogenous levels of tagged EEA1 and APPL1. This requires knock-in the APPL1-EGFP- and TagRFP-T-tagged constructs in the endogenous loci.

We agree that this is the cleanest possible experiment, however given that we observe distributions of APPL1 and EEA1 that match with what has been shown by antibody labelling and CLEM (Van der Beek et al. 2022 *J Cell Biol*), and that we observe APPL1 to EEA1 conversions, we are highly confident that this phenomenology we describe is representative of the endogenous process. These cells are transiently expressing a small amount of fluorescently labelled APPL1 and EEA1 in addition to the endogenous 'dark' proteins. To ensure we were only adding a low amount of additional protein to these cells we used only 30% of total DNA for expressing APPL1 and EEA1 and the rest being blank DNA. When screening for cells we employed two quantitative approaches to ensure we were imaging cells that did not display altered trafficking or conversion dynamics.

1. One of the effects of APPL1 overexpression is enhanced colocalization with Rab5 and EEA1, in agreement with Zoncu et al. 2009 *Cell*. In general, APPL1 and EEA1 only

show very little colocalization in the absence of any artefacts. By co-expressing APPL1 and EEA1 supplemented with blank DNA, we found that only a small percentage of vesicles showed colocalization for APPL1 and EEA1 in a dynamic fashion pertaining to conversions as compared to overexpression, where we see stabilised double positive endosomes.

2. We rely on the direct mean intensity to ensure that we are picking cells that have a very small amount of fluorescently expressing proteins, resulting in minimal perturbation of the processes.

We have discussed these measures to avoid overexpression artefacts in the Methods section on lines 658–662.

3) The authors have designed a pipeline to detect collisions, conversions, and fusions. This pipeline should be thoroughly tested and validated as it is at the core of the experimental paradigm used by the authors in this study. What happens when the routine is used on two vesicular structures that are not expected to interact such as LAMP-1-positive vesicles and APPL1-positive endosomes? I have concerns with the detection of collisions in particular. Two vesicles getting closer to one another do not necessarily “collide” so that they can exchange material. Is there a way to demonstrate transfer of material between two vesicles?

We tested the pipeline thoroughly using three approaches:

1. Firstly, we address the question of collisions versus near misses between nearby vesicles. In order to verify that the number of collisions align with expectations, we performed a back-of-the-envelope calculation using the established method for calculating mean collision frequency in a system containing particles of known physical characteristics. The distributions of endosome displacements (instantaneous velocities), radii (used to calculate effective collision cross-section), and local endosome densities may be used to calculate the theoretical mean time between collisions in this system. The exact measured time between collision is difficult to obtain (due to finite trajectory lengths as a result of occasional tracking errors), but it may be estimated by observing all trajectories lasting up to a specific threshold (here, up to 4 min or 90 frames in length—the longest trajectory length with sufficient data points to plot a distribution). This empirical estimated mean time between collisions sits well within the error bars of the theoretical expected mean time between collisions, thus proving the physical feasibility of our measured collision rates.
2. We followed the reviewer’s suggestion of applying the routine on two vesicular structures that are not expected to interact, APPL1-positive endosomes and late endosomes marked by EGF-Alexa647 at 30 min to 1 h post-stimulation. It is known that EGF traffics through APPL1 endosomes in less than 10 min and subsequently colocalizes with late endosome and lysosome markers (York et al. 2021 *Commun Biol*, Barker et al. 2017 *J Cell Sci*) and thus we would expect these endosomes to be a) less frequently colliding and b) for APPL1 to be unaffected by any collisions. In this case, we detected a smaller number of collisions and zero conversions. We compare the statistics below for APPL1–EEA1 and APPL1–EGF. This also verifies the capability of our analysis code to detect initiation of conversion immediately following collision, as in the case of APPL1-EEA1, EEA1 signals can be detected on tracks of APPL1 masks.

In the case of endosomes bearing EGF, which is found in the lumen, no signal was detected on APPL1 positive endosomes following collisions.

Quantity (x)	Units	$\langle x_{APPL1} \rangle$	$\langle x_{EEA1} \rangle$
Instantaneous velocity	$\mu\text{m s}^{-1}$	0.15 ± 0.11	0.12 ± 0.09
Radius	μm	0.28 ± 0.05	0.42 ± 0.08
Mean number density	μm^{-3}	0.074 ± 0.046	0.030 ± 0.010
Mean free path	μm	7.7 ± 3.3	16.1 ± 6.2
Mean time between collisions; expected	s	45 ± 15	148 ± 72
Mean time between collisions; estimated	s	45 ± 48	49 ± 48

f

	APPL1-EEA1	APPL1-EGF1
Collisions (#)	6953	315
Collisions (# of APPL1 tracks)	14.8%	1.3%
Conversions (# of APPL1 tracks)	2.9%	0.0%

Response Figure 4. Verification of collision detection. Utilising observed distributions of (a) endosome sizes, (b) instantaneous velocities, and (c) local endosome densities, we are able to calculate the expected mean time between collisions; this agrees well with (d) measured mean time between collisions from our data. (e) Averages and standard deviations of corresponding values are provided. The expected mean time between collisions lies well within the error bars of the estimated mean time between collisions, reinforcing that the values we calculate are in line with expectations based on physical principles. (f) Comparison of detected collision and conversion frequencies between APPL1 and EEA1 or EGF-bearing late endosomes.

We also examined the effects of varying the parameters used to count collisions. We varied one of the key parameters used to count collisions, the interparticle distance (reported as the Euclidean distance between the two closest points on the surfaces of nearest neighbours). Up to about 200 nm of separation (Response Fig. 5), the heatmaps of conversion duration versus number of prior collisions show similar trends; above this point, the heatmaps appear more similar to control (nocodazole) data in terms of lack of trend between conversion duration and number of prior collisions, as expected for particles above a certain threshold of separation (that are more likely to not represent true collisions). This separation is roughly in line with measurements from the literature showing that EEA1 extends as a rigid molecule from the endosomal membrane ~200 nm (Murray et al. 2016 *Nature*).

Response Figure 5. Comparison of collision detection analysis parameters. Here we recalculate the heatmaps showing conversion duration versus number of collisions before conversion for various values of the surface-to-surface distance d , as indicated. The trend for values of $d < 200$ nm remains similar, whereas for values of $d > 200$ nm it disappears. Conversions with zero collisions have been excluded from these charts to facilitate comparisons of events where collisions are present.

To provide a better intuition for the meaning of interparticle distances, below we plot the surface-to-surface distance between nearest neighbours for all combinations (APPL1–APPL1, EEA1–EEA1, and APPL1–EEA1). The distinct peak for APPL1–EEA1 at $d < 0$ nm corresponds to cotracking events (only a small fraction of which successfully lead to conversion). Note that the distributions of APPL1–APPL1 and EEA1–EEA1 both peak above $d = 500$ nm, but also show some small overlap at $d < 0$ nm, representing homotypic collisions (APPL1–APPL1 and EEA1–EEA1, respectively).

Distances between nearest neighbours conversions excluded

Distances between nearest neighbours only conversions included

Response Figure 6. Distributions of nearest neighbour distances. The surface-to-surface distance between all pairs of nearest neighbours was calculated, for every combination of channels. There is a distinct bimodal distribution of APPL1–EEA1 distances, indicative of overlap between these channels (only a small fraction of which consists of in-process successful conversions).

4) Collision detection: based on what I can see with the automation procedure, endosomes are given a defined radius. This radius appears quite large to me. Enlargement of endosomes might affect detection of actual collisions. If the radius of endosomes is reduced during the automation procedure, are similar results obtained?

The endosomes are not given a defined radius; rather, the radius is determined using a blob detection routine that computes the Laplacian of Gaussian images with successively increasing standard deviation, at the final iteration returning the standard deviation of the Gaussian kernel that detected the blob. This value is directly proportionate to the approximate blob radius. By visual inspection, this routine resulted in more accurate detection of endosome sizes than alternative approaches (such as pixel-based segmentation). Thus, endosome sizes follow distinct distributions, with the average radius of EEA1 endosomes being 420 ± 80 nm and that of APPL1 endosomes being 280 ± 50 nm.

5) Kinetics of marker intensities (as for example in Figure 1c-d). Intensities of endosomal markers are measured as a function of time in several panels. Is this intensity determined on the same light-sheet plane throughout the experiment or is the light-sheet plane adjusted in in each frame so as to go through the centroid of each endosome analyzed on the image? If the first option was chosen, intensity variations may result from endosomes moving in and out of the plane.

These data were acquired using lattice light-sheet microscopy (LLSM), which uses 2D optical lattices that spread excitation energy across an entire field of view, enabling rapid scanning through the cell to generate 3D volumes (Chen et al. 2014 *Science*). With a depth of focus at 400 nm for a 25x Nikon 1.1 NA objective, and an interplane spacing of 400 nm in our acquisition, there is no ‘out-of-focus’ voxel. All volumes are captured without ‘gaps’. The nature of this imaging modality ensures that the endosome images are always “in the plane”. This is essentially equivalent to the second option suggested. The scan speed over a single endosome (roughly 20 nm/ms) is significantly faster than the velocity of that endosome; thus, intensity variations cannot result from endosomes moving in and out of the plane.

Response Figure 7. Supplementary Fig. 1a.

6) Line 147: “true endosomes”. These “true endosomes” are defined based on fluorescence intensities. Can this be validated somehow? Could the authors use a series of organelle markers (e.g., ER, mitochondria, Golgi markers) and check that their routine does not recognize these as “true endosomes”?

“True endosomes” were initially validated by visual inspection and manual labelling of putative endosomes. Using our selected sets of image analysis parameters, putative endosomes with the highest (lowest) values of intensity were observed to separate into “true” (“false”) endosomes, as defined by visual inspection. Thus, fluorescence intensity of the brightest (dimmest) objects was used as a proxy to define sets of features most similar (dissimilar) to “true” (“false”) endosomes for the purposes of training the machine learning algorithm used to cluster remaining detected objects. Note that analysed movies were subsequently inspected by eye to confirm correct identification of endosomes. In addition, no organelles were observed to be labelled in these data.

7) In Supplementary Figure 3, the terminology used in the workflow is not easy to grasp. It is difficult to understand what terms like “loose tracking criteria” mean as they are not precisely defined. This brings up a general comment: the authors should try to explain their procedures in simpler terms, and they should disclose the actual values of the parameters used in their procedures. Moreover, these parameters should be clearly defined.

In Supplementary Fig. 3a, the step ‘retain events meeting loose tracking criteria’ was used to distinguish the first filtering step (i.e., exclude partial trajectories wherein APPL1 signal did not exist prior to colocalization, and where EEA1 signal did not exist after colocalization) from the third filtering step, labelled ‘retain events meeting strict tracking criteria’ (i.e., exclude

trajectories wherein APPL1 signal was tracked for at least 30 seconds prior to colocalization). The former filter merely ensures sufficient data points before and after colocalization to distinguish conversion events, whereas the latter ensures sufficient points before colocalization to identify putative collision events. We agree these labels are imprecise, and for better clarity have rewritten them and provided full details of parameter values used in new Supplementary Table 1 and updated Supplementary Fig. 3.

8) Figure 4b. Additional controls would be warranted. What is the impact of SAR405 and siRNA INPP4A on PI(3)P levels? How efficient is the siRNA against INPP4A?

We thank the review for their helpful suggestion. We have now added a supplementary figure displaying the levels of PI(3)P as measured by GST-2xFYVE staining following either SAR405 or INPP4A siRNA treatment (Supplementary Fig. 13). In summary, we observe a reduction in both the number of Rab5 endosomes positive for PI(3)P as well as an overall reduction in the amount of PI(3)P detected on Rab5 endosomes, to a level of roughly ~48% and ~40% for INPP4A siRNA and SAR405 respectively. This is referenced in the text on lines 356–360.

9) Figure 6. This model is confusing and does not reflect very well the findings of the authors. In my opinion, the most salient observation is that EEA1 is acquired first by early endosomes through its N-terminal moiety. Then, as endosomes mature, additional EEA1 molecules are recruited via their C-terminal moiety. This is what the model should concentrate on. The other aspects are much more speculative and should be disclosed as such. For example the authors take for granted that collisions allow endosomes to exchange materials but this is not shown in the manuscript. Why is the possibility of unbound, cytoplasmic EEA1 acquisition not considered here? Since APPL1-positive endosomes were observed acquiring EEA1 without collisions, shouldn't this possibility be considered? Why is EEA1 observed binding a single endosome via both its N- and C-terminal domains? It is mentioned earlier on that such interaction is not possible due to the EEA1's rigid quaternary structure. Why is there a full switch between N-terminal and C-terminal EEA1 binding in the model? Why not a gradual acquisition of C-terminal binding on the areas of the endosomes that become PI(3)P positive? How can endosome 1 "steal" EEA1 from endosome 0?

We thank the reviewer for bringing the points of confusion and the questions about the speculations in the model. We agree that some aspects are speculated/logically pieced together based on our findings as well as other findings in relevant literature. We have performed an additional experiment as well as have elaborated on our own data supporting the model in the manuscript as well as have now clearly highlighted the speculative parts.

In brief, we have tracked every single collision in whole cell volume and have followed the intensity of EEA1 on an initially EEA1 'null' endosome. As quantified and reported in the manuscript, 3.6% of APPL1-EEA1 collisions lead to an immediate appearance of EEA1 on APPL1 endosomes. As a negative control, we have now included a new set of experiments where we have performed the exact same analysis on cells expressing APPL1 endosomes and EGF carrying endosomes at 30 minutes and beyond. Because the EGF is in the lumen of the endosome, collision will definitely not result in transfer of fluorescence. We, therefore, validate our analysis that follows an endosome through collision and identifies if the endosome

picks up EEA1 signal in the aftermath of the collision. Secondly, there are a proportion of spontaneous conversions, without any prior collisions, as has been detailed in the Fig. 1f. These are a result of direct binding of EEA1 from the cytoplasm. We have explicitly mentioned these details now in the manuscript on line 194. However, we stress that up to 90% of measured events are preceded by at least one collision or fusion, and correspondingly inhibiting endosomal motility leads to a drastic reduction in the number of measured conversions (Fig. 1e,f and Supplementary Fig. 4).

Due to EEA1's rigid structure it is indeed the case that the majority of molecules bound to an endosome are only able to bind via a single terminus at once, however following binding to endosomal membranes at both ends, EEA1 undergoes an entropic collapse and becomes much more flexible which is essential for its role in endosomal fusion (Murray et al. 2016 *Nature*). Thus, it is highly likely that a proportion of EEA1 molecules may be transiently both N- and C-terminally bound to endosomal membranes before they are re-straightened following fusion, which is represented in the final endosome in the summary figure.

The evidence for a full switch between N- and C-terminally bound EEA1 is as follows. Firstly, based on reduction in FRET efficiency and a corresponding increase in fluorescence lifetime during endosomal maturation, it is consistent with both an increase in the proportion of CT-EEA1 binding and the replacement of NT-bound EEA1 with CT-bound EEA1 (either the same molecules or through replacement with EEA1 dimers from the cytoplasm). This is also visualisable in the phasor plots of EEA1-EGFP and Rab5-RFP expressing cells (Response Fig. 8). The phasor plot distribution of pixels from perinuclear endosomes closely matches the phasor plot of EEA1-EGFP (donor) only cells. In comparison peripheral endosomes match the phasor plots observed in SAR405 treated cells which have minimal C-terminal EEA1 binding due to abrogation of VPS34 production of PI(3)P.

Response Figure 8. Summary of phasor plots of EEA1 EGFP cells. Top, phasor plot of EEA1 EGFP only cells showing the EGFP lifetime in the absence of RFP acceptor FRET. Middle, EEA1 EGFP Rab5 RFP expressing cells treated with SAR405, leading to a majority of molecules to FRET. Bottom, phasor plot of untreated EEA1 EGFP Rab5 RFP expressing cells with donor pixels separated into whether they are from perinuclear (left) or peripheral (middle) endosomes. Bottom right plot shows donor and acceptor lifetimes of all endosomal pixels.

With regards to one endosome ‘stealing’ another’s EEA1, it is important to highlight that at the single molecule level these endosomal proteins are dynamic, and frequently show stochastic binding and unbinding (see Fig. 1h and Supplementary Fig. 5, as well as the discussion of stochasticity above). From our analysis routine we observe a subset of APPL1 endosomes acquire EEA1 signal immediately post collision (Supplementary Movie 3, Supplementary Movie 6). In comparison, in our control experiments where we have detected collisions between APPL1 and labelled EGF, we never observe a transfer of signal (Response Fig. 4).

In line 389, the authors state that “collisions between endosomes form an important step in overall endosomal conversions rates”. I do not think the data permit reaching this conclusion. As indicated above, characterizing collisions is prone to many caveats. There may be a correlation between endosomes having experienced collisions but whether these putative collisions increase the conversion rate is one possibility among others. One could argue that it is the velocity of the endosomes through their interactions with microtubules and motor proteins that facilitate their conversion, not the bouncing of endosomes on other vesicles (faster moving endosomes are more likely to encounter other vesicles but these encounters may play no role in endosome maturation). In conclusion, I feel that the authors should tune down some of their claims and, in their model, concentrate on their most solid findings.

We disagree with the reviewer that the detection of collisions is prone to many caveats and we hope that we have satisfactorily addressed the concerns raised and demonstrated with the additionally included data that we are able to robustly track and detect collisions.

Large amounts of collision-conversion events are detected even with the most stringent criteria of physical interactions and the intensity thresholds set for conversion. As shown in Fig. 1f only 12% of APPL1 endosomal conversions occur spontaneously, with the bulk of conversions either occurring following a heterotypic collision, collision followed by a rapid fusion or via a direct fusion. All of these processes rely on the molecular interactions of EEA1 to bind to endosomal membranes at both the N- and C-terminus, and through subsequent maturation processes leads to a tight distribution of conversion times. This is further supported by our simulations (Fig. 5) which highlight the requirement of large influxes of N-terminally bound EEA1 (occurring through collisions or fusion with EEA1 endosomes) for timely conversions and for a tight distribution of first passage conversion times.

Minor

1) I would refrain from using normative words that are poorly meaningful in biology such as “robust”.

In this manuscript we utilise the term ‘robust’ or ‘robustness’ as it is used in the fields of biophysics and dynamical systems, to refer to systems and behaviours that are robust to perturbations or fluctuations. Please see Lesne 2008 *Biol Rev* for a discussion on the concept of robustness as it pertains to biology.

2) What is the usefulness of mentioning dynein in the introduction and raising questions related to this protein as the paper is not directly addressing the role of dynein in endosomal maturation?

The link between movement in intracellular localisation and biochemical maturation until now has been clarified on lines 62–70. It has been shown in previous work by the lab that inhibition of dynein in the context of growth factor trafficking leads to impaired endosomal maturation which motivated this study of endosomal timekeeping mechanisms (York et al. 2021 *Commun Bio*).

3) Line 128: “To simultaneously measure ensemble endosomal conversion dynamics”. I do not understand this sentence.

This sentence has been updated: ‘To both measure ensemble endosomal conversion dynamics, as well as ...’.

4) Supplementary movie 4. This movie should be “labelled” so that we can follow the authors’ interpretation. For example, in the control cells, where are all the conversion events? It would be nice to have them indicated by labels.

We have now annotated Supplementary Movie 4 as suggested.

5) Line 208: “were also “pulsatile” suggesting evidence of clustering”. What is the rationale of this statement? Couldn’t this “pulsatility” due to vesicles moving in and out of the light sheet?

The appearance or disappearance of large intensity signals corresponding to many molecules is evidence of binding or unbinding of clusters of molecules. It is not possible for molecules to move “in and out of the light sheet” as the light sheet is scanned across the volume of the cell at roughly the rate of 20 nm/ms which is much faster than the speed at which endosomes move.

6) Line 219: supplementary figure 6. To which structures are all the little dots corresponding to?

These small dots are most likely individual non-membrane bound molecules, although there may also be minor contributions from spurious non-specific signals.

7) Line 224 and 226: the figures are not supplementary.

Corrected.

8) Line 257: Fig. 2c. We do not see the nucleus in the figure. Please label it.

We have now updated the figure legend to improve clarity. Given that we have not stained specifically for the nucleus, we believe it would be inappropriate to label an approximate position. In this context we reference perinuclear and peripheral endosomes to distinguish between endosomes at the edge of the cell and the remaining majority of endosomes. The use of terms perinuclear region and perinuclear endosomes is consistent with the bulk of the literature within the endosomal field.

9) Line 349: Supplementary Figure 9. This figure lacks a control (untreated cells)

Apologies, this is now included.

10) Is there a repository available for all the collected data used in the article?

To our knowledge the largest accessible repositories currently allow around 50 GB to be stored, unfortunately the total data collected for this article is around 100 TB; we are looking into making some representative datasets available but will have to keep the bulk of it locally and make it available upon request.

11) Supplementary Figure 5c. I do not understand this figure. What does it show? How can it "hint at periodicity"?

In Supplementary Fig. 5c, we represent the data using the autocorrelation function, which represents the correlation of a time series and its lagged version over time. In other words, it shows the similarity between observations of a random variable made at different times (defined by the time lag or interval between observations). The analysis of autocorrelations is a mathematical tool for finding repeating patterns, especially periodic signals that may be obscured by noise. In our case, the autocorrelation function does not monotonically decrease to zero, but rather exhibits peaks and valleys, indicating the presence of periodicity in the observed variable.

12) Figure 1g. The clustering of APPL1 and EEA1 should be quantitated somehow. At present, only one frame is shown and only one vesicle was analyzed over time.

We have quantified (counter-)clustering of APPL1 and EEA1 in a complete movie (200 frames) by identifying the signal pixels in each respective channel, then calculating the percent of total pixel signals (either APPL1 or EEA1) displaying overlap (both APPL1 and EEA1) across the movie. Response Fig. 9 has been added as Supplementary Fig. 8.

Response Figure 9. Quantification of counter-clustering. (a) Sample SRRF image showing APPL1 (cyan), EEA1 (magenta), and APPL1 plus EEA1 (yellow). (b) Binary mask corresponding to a, following thresholding to identify clusters of signal in each channel, showing pixels identified as APPL1 (cyan), EEA1 (magenta), and APPL1 plus EEA1 (yellow). (c) Percentage of overlapping signal pixels per frame, calculated as total pixels containing both APPL1 and EEA1 signals divided by total pixels containing APPL1 and/or EEA1. In any given frame, the overlap of segmented APPL1+EEA1 channels, as a percentage of total signal pixels present in either the APPL1 or EEA1 channel, is $13.1 \pm 3.7\%$, indicating that approximately 87% of all clusters contain either detectable APPL1 or EEA1 signal, but not both. (d–f) Volume renderings of segmented images over 200 frames, showing (d) APPL1, (e) EEA1, and (f) APPL1+EEA1. APPL1 signal is indicated by cyan, EEA1 by magenta, and overlap (APPL1+EEA1) by yellow shading. Time is represented along the vertical axis.

13) Figure 2c. Was the RFP signal obtained solely through FRET or was RFP signal obtained after illuminating the RFP fluorophore with the RFP excitation wavelength?

The RFP intensity channel was acquired using 560 nm excitation and collection of photons between 580–630 nm.

14) In Supplementary movie 6, shouldn't we see the blue signal (N-terminally-bound EEA1) eventually being replaced by the green signal (C-terminal-bound EEA1)? This does not appear to be the case in this movie.

In this movie, we zoomed into a peripheral region of the cell in which we are observing at fast timescales the initial collision events and first EEA1 binding. The NT to CT conversion takes place over a longer period of time (the time axis in Fig. 3e is over 1000 s); typically this occurs coincident with motion and other phenomenology. An example N- to C-terminal conversion is included in the following movie (Supplementary Movie 7).

15) Figures 4b and 4d. Please describe precisely how the boxplots were constructed. What do they show? Same comment for the violin plots. What are the diamonds in the violin plots?

In both Fig. 4a and 4d, the raw measured data is shown as circles or diamonds. The violin plot corresponds to a normal distribution of all events. The box plot corresponds to the 25–75 percentile events with the bars showing total range. The means are depicted by the open squares. We have now added this to the figure legend in the manuscript.

16) Line 380: “the localisation of this [R1375A] mutant was largely cytosolic”. Isn't this unexpected as the first binding occurs via the other end (the N-terminus)? This mutant should therefore be seen on very early endosomes.

Given that the N-terminus of EEA1 has a relatively weak binding affinity for Rab5 (Dumas et al. *Molecular Cell*, 2001; Merithew et al. *JBC*, 2003) that is lower than the affinity of APPL1 for Rab5 and PI(3,4)P2, we only observe transient binding as reported earlier (Gaulhier et al. *JBC*, 2005) and a majority of fluorescence signals is cytosolic in live cells. Upon fixation, we do see N-terminal mediated binding as the reviewer points out (supplementary figure 9d).

17) There is an inconsistency between what is stated in line 430 (“the N-terminus of EEA1 has weak binding affinity to Rab5”) and in line 482 (“The N-terminus forms the stronger Rab5 interaction”).

We thank the reviewers for pointing out this inconsistency. In fact, what we meant was the N-terminus forms the stronger Rab5 interaction compared to only Rab5 interaction of C-terminus. The C-terminal interaction, owing to the FYVE domain, operates on a coincidence detection mechanism resulting in specificity to PI(3)P-containing vesicles. However in the presence of PI(3)P, the FYVE domains at the C-terminus of EEA1 lead to strong association with Rab5-PI(3)P endosomal membranes by coincidence detection, stronger than the N-terminus Rab5 interaction (Carlton and Cullen 2005 *Trends Cell Biol*). We have now rephrased it for clarity lines 523–528.

18) Figure 1b. Doesn't b fuse with a? If not where is a after the “collision”. In this figure, it would be help the readers that endosomes a to d are indicated on all the images where they

are found (from their initial detection to their eventual disappearance). This comment is valid for other similar figures where endosomes are labelled.

In this figure, there is a collision between 'a' and 'b' in the first frame, where-after one of them exits the field of view before the next selected frame (37 s later). We updated the labelling of these endosomes to be more clear, and note in the figure captions how the endosomes are labelled.

19) Figure 1d (legend). Typo: "endosome" not "endsome".

Corrected.

20) Figure 4b (graph). For clarity I suggest to write "APPL1 to EEA1 conversion" and "Fusion between APPL1 and EEA1 vesicles" directly on the figure.

This has now been clarified in the figure legend.

Reviewer #3 (Remarks to the Author):

This manuscript used a battery of cutting-edge, fluorescence based quantitative imaging approaches to build a model of very early endosome maturation.

Lattice light sheet microscopy allows complete volumetric imaging of RPE cells expressing fluorescent APPL1 and EEA1 endosomal proteins at diffraction-limited and ~second time intervals over extended periods (many minutes). Combined with an unbiased, automated segmentation and tracking pipeline, the manuscript is able to produce a pan-optical reconstruction of the maturation process, which produces the first surprise in how maturation occurs: although the authors do observe occasional spontaneous conversion of APPL1 endosomes into EEA1 endosomes (12%), these are usually preceded by one or more collisions with EEA1 endosomes (11%) or else occur by fusion with existing EEA1 endosomes (77%). Using FLIM of an N-terminally tagged EEA1 with a tagged Rab5, the manuscript can differentiate EEA1 bound to Rab5-positive endosomes via their N- or C-termini. Curiously, fusion is only observed to occur between EEA1 labelled endosomes where at least one endosome is attached by C-terminally anchored EEA1, whereas maturation involves a transition from N-terminally to C-terminally dominated EEA1. The endosomal lipid PI(3)P is predominantly associated with c-terminally attached EEA1 as expected. Inhibition of the major PI(3)P-synthesizing VPS34 enzymes abolished endosome fusion, but not conversion - whereas blocking PI(3,4)P2 to PI(3)P conversion using INPP4A siRNA blocks both processes - implying a PI(3,4)P2 to PI(3)P switch is essential to both modes of maturation. An N-terminal mutant of EEA1 unable to bind Rab5 mildly inhibits fusion but not conversion in RPE1 cells, but cannot restore maturation at all in EEA1 KO HeLa cells. The observations are synthesized into a unifying model of endosome maturation: an initially PI(3,4)P2, Rab5 and APPL1 coated very early endosome collides with c-terminally anchored EEA1 endosome, which allows n-

terminal binding of Rab5. This interaction displaces APPL1, allowing further EEA1 binding. This cascades until such a time that enough APPL1 has been displaced to free Rab5 binding sites and for INPP4 to convert the lipid to PI(3)P, which allows either maturation by c-terminal binding of EEA1 or else fusion with more mature endosomes.

Overall, the manuscript is technically impressive and produces a novel model for endosome maturation that is still by and large consistent with prior literature. The manuscript is clearly written, and the data are beautifully presented and convincing. I believe the manuscript will be an important vertical advance in the field of membrane traffic, with the approach and concepts applicable to other trafficking processes. The manuscript is therefore of significant interest to a wide swathe of the biological and biomedical sciences, and appropriate for publication in Nature Communications (or Nature Cell Biology, honestly - that would be a better home in my opinion).

We thank the reviewer for the kind comments and have strived to address all of his concerns below.

Overall, the reviewer found no major flaws. However, there is one area that could be improved. Specifically, the sections describing “Endosomal conversions are driven by phosphoinositide conversions by INPP4A” seemed overly simplistic to me:

Firstly, although SAR405 barely inhibits conversion of VEEs compared to INPP4A siRNA, it is rather more effective at blocking fusion. However, this is not commented on in the text, despite the fact that it would make sense as PI3P is more prevalent on the more mature, C-terminally EEA1-bound early endosomes. Fourthly, the fact that INPP4A inhibition and VPS34 inhibitors blocks fusion is curious and unexpected; it implies that lipid is required on both endosomes (VPS34-derived on later early endosomes, and INPP4A-derived on VEEs). However, this is not addressed in the manuscript.

We thank the reviewer for pointing this out. We have now included discussing this point in the manuscript in lines 568-577. In the case of SAR405 treated cells, PI3P production from PI is impaired, resulting in PI3P deficient mature Rab5 positive endosomes. APPL1 endosomes can still undergo conversion because of PI3P production from INPP4A but is impaired in fusion owing to a substantial decrease in the number of C-terminally bound EEA1 mature vesicles in SAR405 treated cells (supplementary fig. 14).

In the case of INPP4A siRNA treatment, PI3P production on initially APPL1 positive endosomes is impaired, contributing to a decrease in fusion. Secondly, conversion is entirely abrogated, that leads to a drop in the overall levels of PI3P positive mature endosomes. These two factors contribute to lack of fusions. As the reviewer points out, this hints towards requirement of C-terminally bound EEA1 on both endosomes, however, as measured by our FRET experiments in untreated cells, it suffices to have one of the endosomes C-terminally bound (Fig. 3c). In agreement with this, we do see some level of fusions in SiRNA INPP4A treated cells (~20% of untreated wild-type cells).

Secondly, the manuscript lacks clear controls for the extent of PI(3)P depletion by these two treatments. Data should be included documenting the extent of PI(3)P depletion by 100 nM SAR405 or INPP4A knock-down, using the GST-2xFYVE probe utilized in other experiments.

We thank the review for their helpful suggestion. We have now added a supplementary figure displaying the levels of PI(3)P as measured by GST-2xFYVE staining following either SAR405 or INPP4A siRNA treatment (Supplementary Fig. 13). In summary, we observe a reduction in both the number of Rab5 endosomes positive for PI(3)P as well as an overall reduction in the amount of PI(3)P detected on Rab5 endosomes, to a level of roughly ~48% and ~40% for INPP4A siRNA and SAR405 respectively. This is referenced in the text on lines 359–363.

Thirdly, PI(3,4)P₂ conversion to PI(3)P being the source of this lipid required for the conversion process is intriguing, and could imply that the lipid ultimately comes via the PI3KC2A pathway; this can now be directly tested with highly potent and selective PITCOIN inhibitors of these enzymes (doi: 10.1038/s41589-022-01118-z). It would be interesting and informative (but not essential) to this model to test whether these inhibitors also block conversions and fusions.

We thank the reviewer for the suggestion; whilst it is indeed interesting it appears that these inhibitors are currently not yet easily available. However, this is an avenue for further study.

Fourthly, the fact that INPP4A inhibition and VPS34 inhibitors blocks fusion is curious and unexpected; it implies that lipid is required on both endosomes (VPS34-derived on later early endosomes, and INPP4A-derived on VEEs). However, this is not addressed in the manuscript.

Answered above.

Lastly, the model discussed at the end of the manuscript describes a direct interaction of APPL1 with PI(3,4)P₂. However, my recollection of the literature is that this is essentially implicit from the co-incident timing of effectors and PI enzymes, coupled with poorly defined lipid binding by APPL1. However, I cannot recall explicit demonstrations of a specific interaction between APPL1 and PI(3,4)P₂. If there is one, this should be stated; if not, the implicit nature of the interaction should be spelled out in the manuscript.

Chial et al. 2008 *Traffic* show that APPL1 recognises both PI(3,4)P₂ and PI(3,4,5)P₃ via its PH domain using protein overlay experiments coupled with systematic deletions; we have now made specific mention of this fact in the manuscript.

I also spotted a couple of places where technical clarifications are required:

Figures 2b, 3a, b and d: it is not always clear what the data refer to, in terms of individual data points, box and whisker or violin plots. How do these relate to the number of endosomes, number of cells, and number of experiments? This information should be included in the

legends. Likewise for the summary statistics included in the box plots - its not clear if these are means or medians etc.

We thank the reviewer for bringing this to our attention. We have now clarified the appropriate figure legends with the missing information.

Reviewers' Comments:

Reviewer #1:

Remarks to the Author:

Overall the authors have answered all my queries and questions satisfactory.

Minor

In the particular case of the graphs where the lifetime and intensity are shown (Response Figure 3) Are the plots representing pixel-based information? Perhaps it would be more representative to show the data integrating both the intensity and the lifetime for individual endosomes.

In Response figure 2, the authors show a lifetime histogram comparing the lifetimes of EEA1 Rab5 endosomes (peripheral and peri nuclear) to demonstrate that fixation protocols did not affect the lifetimes. These distributions show an error (half width of the distribution) of around 0.5 ns for the ensemble of FLIM data (not sure if this is integrated per endosome or just shows the pixel by pixel distribution). In principle this matches well with Response Figure 3f. The co-expression of EEA1-GFP and Rab5-RFP to check for FRET shows standard deviations of 0.5 to 0.3 ns. However the differences in lifetime when FRET occurs are only around 0.1 ns. Even if FRET seems to be independent of donor/acceptor ratios (pixel by pixel...) the FRET diminution is too short given the error of the measure. Please comment.

Finally, the authors explain that when expressing their labelled proteins they do so in little amounts so that endosomal trafficking is not disrupted. Clearly the best way to approach this would be generating CRISPR-Cas9 cells expressing endogenous amounts of EEA1-GF and Rab5-RFP. Perhaps to validate this point it would be worth of performing an experiment with fixed cells employing labelled antibodies against EEA1 and Rab5. Since the authors have elegantly demonstrated that one can conduct FRET FLIM in fixed cells.

Reviewer #2:

Remarks to the Author:

I thank the authors for the care taken to answer my queries. I am now globally satisfied with their answers. Because it shows validation of the authors' approach, I am suggesting that their answer to my query #3 is added to the manuscript as supplemental data (Response Figure 4 to 6).

Reviewer #3:

Remarks to the Author:

Overall, the revised manuscript addresses my major concerns with the original. I believe the manuscript in its current form is suitable for publication, with a couple of minor changes that can be addressed with modification to the text:

(1) It is understandable that PITCOIN experiments are not yet feasible with these non-commercial compounds. However, it is worth citing this study (doi: 10.1038/s41589-022-01118-z) in the context of the experiments presented in figure S13, since PITCOINs have similar effects on PI3P levels as INPP4A siRNA in this study.

(2) I am still unconvinced that APPL1 selectively interacts with PI(3,4)P2. The Chial et al paper shows rather non-selective interactions of both PTB and PH domains with phosphoinositides. Whilst a PI(3,4)P2 interaction with APPL1 on VEEs is still the most likely explanation for the current data, I think that the interaction should not be reported as established fact.

REVIEWERS' COMMENTS – our responses are in blue

Reviewer #1 (Remarks to the Author):

Overall the authors have answered all my queries and questions satisfactory.

Minor

In the particular case of the graphs where the lifetime and intensity are shown (Response Figure 3) Are the plots representing pixel-based information? Perhaps it would be more representative to show the data integrating both the intensity and the lifetime for individual endosomes.

In Response figure 2, the authors show a lifetime histogram comparing the lifetimes of EEA1 Rab5 endosomes (peripheral and peri nuclear) to demonstrate that fixation protocols did not affect the lifetimes. These distributions show an error (half width of the distribution) of around 0.5 ns for the ensemble of FLIM data (not sure if this is integrated per endosome or just shows the pixel by pixel distribution). In principle this matches well with Response Figure 3f. The co-expression of EEA1-GFP and Rab5-RFP to check for FRET shows standard deviations of 0.5 to 0.3 ns. However the differences in lifetime when FRET occurs are only around 0.1 ns. Even if FRET seems to be independent of donor/acceptor ratios (pixel by pixel...) the FRET diminution is too short given the error of the measure. Please comment.

Finally, the authors explain that when expressing their labelled proteins they do so in little amounts so that endosomal trafficking is not disrupted. Clearly the best way to approach this would be generating CRISPR-Cas9 cells expressing endogenous amounts of EEA1-GF and Rab5-RFP. Perhaps to validate this point it would be worth of performing an experiment with fixed cells employing labelled antibodies against EEA1 and Rab5. Since the authors have elegantly demonstrated that one can conduct FRET FLIM in fixed cells.

At ensemble levels of endosomes, pooled from various randomly shaped cells, results in difficulties in precisely defining peripheral and perinuclear endosomes; due to this, the differences in lifetime become minimal, because they are averaged over poorly defined category of endosomes. However, one needs to follow the endosome through time to see the N-terminal to C-terminal shift clearly or observe them individually in slightly elongated cells (Fig. 2C), that allows distinguishing them more precisely. As in the below example image, within the same region, one can see a mix of endosomes with low intensity ratio-low lifetime (endosome #1), high intensity ratio-high lifetime (endosome #2), low intensity ratio-high lifetime (endosome #3), and high intensity ratio-low lifetime (endosome #4). This mix is difficult to distinguish very clearly in an description of peripheral vs perinuclear which itself is arbitrary resulting in apparent low differences. However, when an individual endosome is followed, the change in lifetime >0.4 ns is clearly observed. The distinction is also clear when elongated cells are selected, where perinuclear and peripheral endosomes can be clearly discerned.

Using antibodies, even if specific towards the N- or C- terminal (Murray et al. Nature, 2016) will not work for fluorescence lifetime owing to the size of the antibodies. We thank the reviewer for their input and will endeavour to conduct future studies with generation of CRISPR-Cas9 cells. Given that there is a clearly observable over-expression artefact that we can avoid as well as we select cells for very low expression levels based on intensity, we believe our observations are valid. Further such approaches have been used for mapping phosphoinositides or endosomal populations pertaining to APPL1 and associated endosomes etc. (He *et al.* Nature, 2017, Sposini *et al.* Cell reports, 2017, Goto-Silva *et al.* Scientific Reports, 2019).

Reviewer #2 (Remarks to the Author):

I thank the authors for the care taken to answer my queries. I am now globally satisfied with their answers. Because it shows validation of the authors' approach, I am suggesting that their answer to my query #3 is added to the manuscript as supplemental data (Response Figure 4 to 6).

We thank the reviewers for their valuable inputs and have now added the details of pipeline to supplemental data.

Reviewer #3 (Remarks to the Author):

Overall, the revised manuscript addresses my major concerns with the original. I believe the manuscript in its current form is suitable for publication, with a couple of minor changes that can be addressed with modification to the text:

(1) It is understandable that PITCOIN experiments are not yet feasible with these non-commercial compounds. However, it is worth citing this study (doi: 10.1038/s41589-022-01118-z) in the context of the experiments presented in figure S13, since PITCOINs have similar effects on PI3P levels as INPP4A siRNA in this study.

We thank the reviewer for the input and have now added this.

(2) I am still unconvinced that APPL1 selectively interacts with PI(3,4)P2. The Chial et al paper shows rather non-selective interactions of both PTB and PH domains with phosphoinositides. Whilst a PI(3,4)P2 interaction with APPL1 on VEEs is still the most likely explanation for the current data, I think that the interaction should not be reported as established fact.

We have amended the sentence to reflect it as an interpretation than an established fact.